# Feihu Dong, a New Hotspot Cave of Subterranean Biodiversity from China

**Sunbin Huang** [1,2], **Mingzhi Zhao** [1], **Xiaozhu Luo** [1], **Anne Bedos** [3], **Yong Wang** [4], **Marc Chocat** [5], **Mingyi Tian** [1] and **Weixin Liu** [1,*]

1   Department of Entomology, College of Plant Protection, South China Agricultural University, 483 Wushan Road, Guangzhou 510642, China; huangsunbin@163.com (S.H.); zhaomzhai@gmail.com (M.Z.); xiaozhu.luo@scau.edu.cn (X.L.); mytian@scau.edu.cn (M.T.)

2   Henry Fok School of Biology and Agriculture, Shaoguan University, 288 Daxue Road, Shaoguan 512005, China

3   Institut de Systématique, Évolution, Biodiversité (ISYEB), UMR7205, CNRS, Muséum National d'Histoire Naturelle, Sorbonne Université, EPHE, 45 rue Buffon, 75005 Paris, France; bedosanne@yahoo.fr

4   Xiangxi Cave Expedition, 27 Jianxin Road, Jishou 416000, China; wangyong19711018@163.com

5   SHAG Caving Association, 24 rue des Roses, 25000 Besançon, France; mchocat@aol.com

\*   Correspondence: da2000wei@163.com

**Abstract:** China is a country with abundant karst landscapes, but research on cave biodiversity is still limited. Currently, only Ganxiao Dong, located in Huanjiang, Guangxi, has been reported as a hotspot for cave biodiversity. Many of the world's most troglomorphic species in the major groups of cave animals have been recently discovered in China, making the existence of many more hotspots in the country likely. Feihu Dong, one of these potential hotspot caves, has been systematically investigated to complement a preliminary species list of 1995, leading to the discovery of 62 species of animals from the cave. Among them, 27 are considered troglobionts or stygobionts, 26 are considered troglophiles or stygophiles, and nine are classified as trogloxenes or stygoxenes. Research on the cave biodiversity of Feihu Dong has demonstrated that it currently holds the highest number of known cave animal species in China. Among the most remarkable features of this fauna is the co-occurrence of five species of cave-obligate beetles, all modified for cave life. The biological survey was limited to a small part of the cave. Several habitats (like guano) have not been investigated so far, and several important cave groups have been insufficiently or not sampled (like Ostracoda). Meanwhile, the system increases in length with each new caving expedition. Further discoveries of cave organisms in Feihu Dong are therefore expected. As Feihu Dong and Ganxiao Dong are the only caves in China that have been extensively studied for a large range of organisms, and as they are located in karstic areas that are similar in richness to other regions of southern China, it can be confidently assumed that several other caves of high biodiversity will be discovered in the coming years.

**Keywords:** South China Karst; Hunan; Wulongshan; cave fauna; stygobionts; troglobionts; diversity; checklist; conservation

## 1. Introduction

China has the largest karst area in the world, covering 3.4 million km² of soluble rock, including an exposed carbonate rock area of 910,000 km² [1–3]. The South China Karst, one of the world's largest and most geomorphologically diverse wet tropical-subtropical karst landscapes, stretches from the Qinling Mountains in the north to the Guangxi Basin in the south and from the Hengduan Mountains in the west to the Luoxiao Mountains in the east. This karst crosses the three-step terrain in China from west to east, with an

elevation of 110–2100 m and an area of 550,000 km[2] [4]. It contains seven exceptional karst landscape clusters that have been designated World Heritage Sites [5].

China not only presents stunning karst landforms but also abundant caves and an exceptional cave biodiversity [6]. It is estimated that there are more than 500,000 caves in China [2]. Currently, the longest explored cave system is Shuanghe Dong in Suiyang County, Guizhou Province, with a reported length of 257.4 km in 2021 [7], which has since been extended to 400.8 km with connected caves increasing from 64 to 105 (Qian Z., pers. comm.). The cave with the largest chamber volume is Miaoting, located in Getuhe, Guizhou Province, with a volume of 10.78 million cubic meters [8]. China is also home to many beautiful and huge caves and tiankengs (the term for giant dolines), such as Zhijin Dong in Guizhou, Shui Dong of Benxi, Liaoning, Shihua Dong in Beijing, or the tiankengs of Dashiwei in Guangxi and Xiaozhai in Chongqing [9,10].

In recent years, the descriptions of subterranean species new to science has considerably increased [11–17]. To date, only Ganxiao Dong, situated at the junction of the Mulun Karst in Guangxi and the Maolan Karst in Guizhou, has been reported as a regional hotspot of cave biodiversity, with 26 species of cave invertebrates reported, including 20 species of troglobionts and six species of troglophiles [18]. However, our investigation indicates that numerous other areas in China also have a high number of cave species, such as Huoyan Karst in northwestern most Hunan, Hanzhong Karst in southern Shaanxi, Du'an Karst in Guangxi, and Wulong Karst in Chongqing, among others [19].

Wulongshan National Geopark is located in Longshan County, Xiangxi Tujia and Miao Autonomous Prefecture, Hunan Province, at the junction of Hunan and Hubei Provinces and Chongqing Municipality. The most famous scenic spots in Wulongshan Park are in the Huoyan Karst, where 212 caves have been recorded [20], including Feihu Dong, Wulong Dong, Shihua Dong, Feng Dong, and Lianyu Dong. Among them, Feihu Dong is the most spectacular. From 1993 to 2002, cave explorers and biologists from the Sino-French joint expedition team and from other countries such as Slovenia, Belgium, and Japan came to Feihu Dong for several scientific expeditions [21–23]. The cave is today nearly 20 km long and has not been fully explored (Figure 1). Underground rivers, lakes, boulders, and side passages have been reported in Feihu Dong [22,24]. The cave is inhabited by abundant and diversified subterranean fauna that includes a number of troglobiotic/stygobiotic and endemic species [25–27], such as *Triplophysa xiangxiensis* (Yang, Yuan and Liao, 1986); *Caridina longshan* Cai and Ng, 2018; *Toshiaphaenops ovicollis* Ueno, 1999; *Angustopila huoyani* Jochum, Slapnik and Páll-Gergely, 2014 [23,28–30].

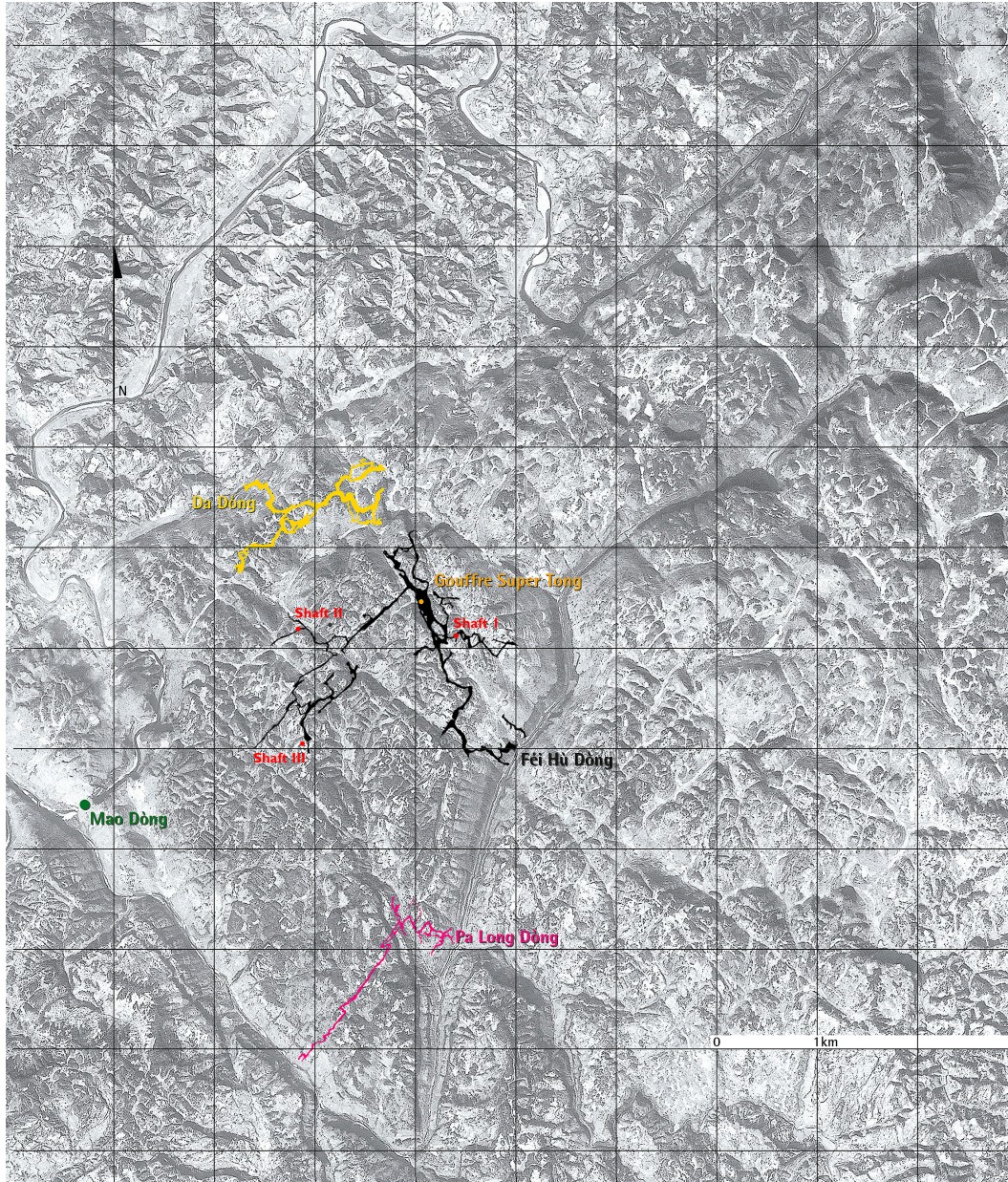

**Figure 1.** Overview of the geographical location of Feihu Dong and other caves in Huoyan Karst: base map and data from SHAG Caving Association, except the location data of three potential shafts from the Xiangxi Cave Expedition.

After the joint expeditions of 1993–2002, there has not been any further systematic cave survey or biological sampling in Feihu Dong. Recently, in February 2023, an exploration project of Feihu Dong was resumed under the leadership of the Caves Committee of the Geological Society of China. The project was mainly executed by the Xiangxi Cave Exploration Team, which used paperless cave surveying and 3D laser scanning technology to re-explore and investigate Feihu Dong. In April 2023, the team have completed the re-survey of approximately 4 km from the main entrance and the newly discovered approximately 2.2 km of cave passage during the last three explorations (Wang Y., pers. comm.). At the same time, the South China Agricultural University (SCAU) biocaving team conducted a week-long systematic survey of cave fauna in Feihu Dong in February 2023 and discovered additional cave animals.

The purpose of this study is to provide an updated list of animals living in Feihu Dong, to draw attention to the scientific importance of these species and their fascinating habitats, and to contribute to the subterranean biodiversity in China.

## 2. Materials and Methods

### 2.1. Research Site

Feihu Dong (飞虎洞, "Flying Tiger Cave" in Chinese) (Figure 1) is located in the Grand Canyon of Wulong Mountain, in Longshan County, Xiangxi Tujia and Miao Autonomous Prefecture, northeastern Hunan Province.

Feihu Dong is a complex cave system (Figure 2) with three entrances (Figures 3 and 4A–D), one of which is a −320 m deep shaft (Figure 3), long galleries (Figure 4E) adorned with speleothems (Figure 4F,G), large chambers, subterranean lakes and rivers. The length of the explored and surveyed cave passages is about 20 km in total. The main entrance of the cave, Feihu Dong (coordinates: 29°12′28.4″ N 109°18′16.4″ E), is at an altitude of around 360 m a.s.l. A three-kilometer-long gallery from the entrance is connected to a large chamber of 26,400 m² in surface. A large shaft named "Gouffre Super Tong", with a waterfall inside, opens on the karst surface and leads 320 m deeper to the northern part of this chamber (Figure 3). In addition, three potential shafts that may be connected to Feihu Dong are marked on the map (Figure 1) (Wang Y., pers. comm.). Some location names within Feihu Dong are translated between French, English, and Chinese in Table 1.

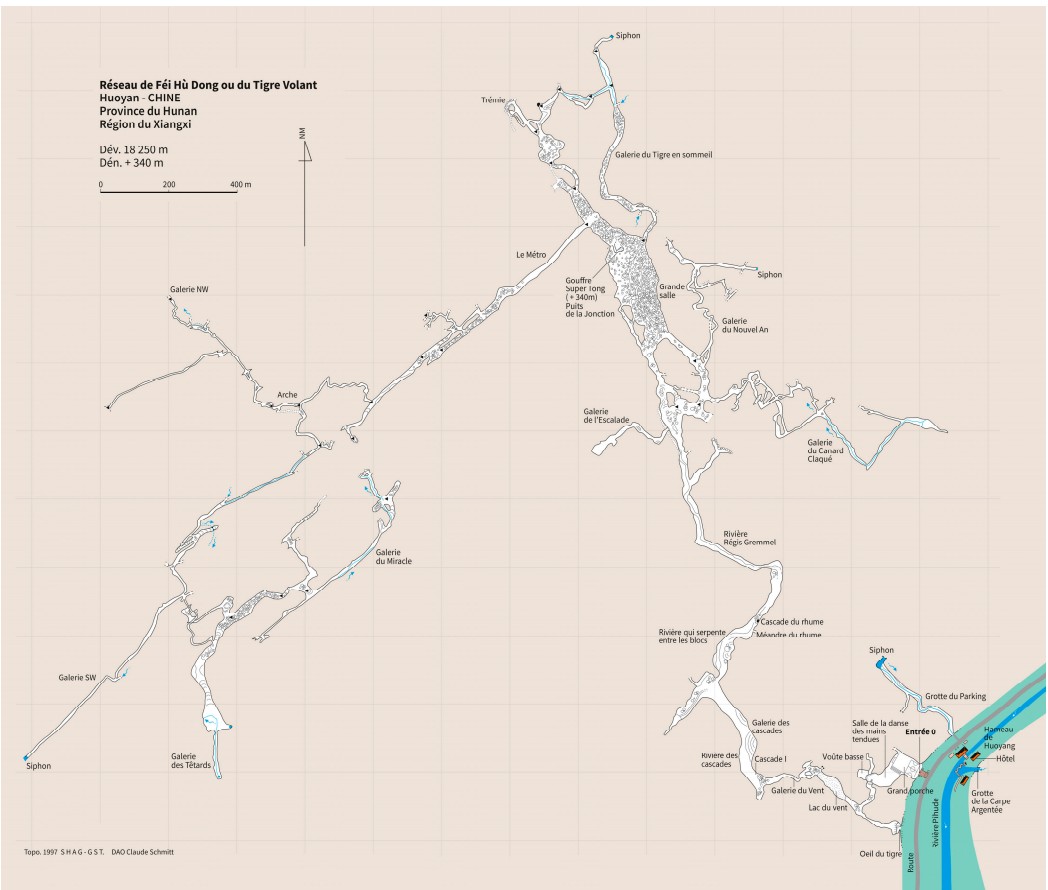

**Figure 2.** Cave map of Feihu Dong: "Topographie de la Grotte du Tigre Volant (Feihu Dong)" drawn by SHAG Caving Association.

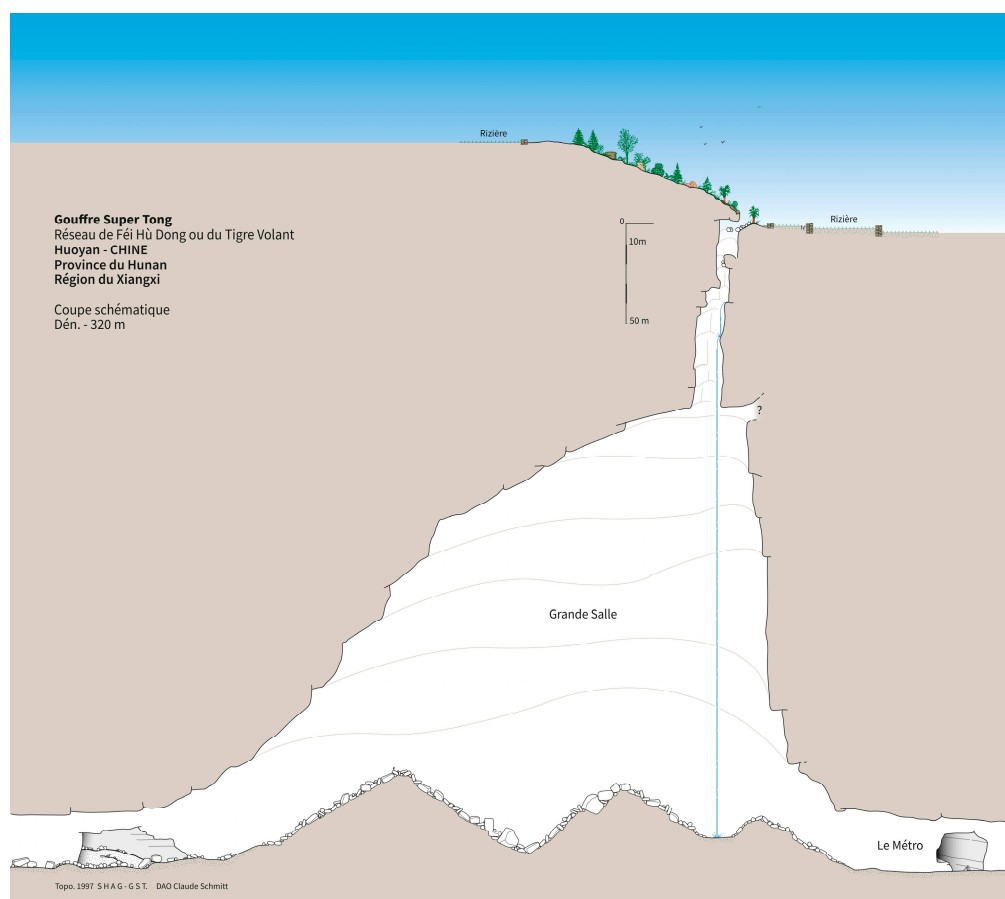

**Figure 3.** Cave map of the shaft "Gouffre Super Tong", explored and mapped by the SHAG Caving Association.

**Table 1.** Names of some locations in Feihu Dong.

| French | English | Chinese |
|---|---|---|
| Grand porche | Large Porch Entrance | 洞口 |
| Oeil du tigre | Tiger Eye | 虎眼 |
| Salle de la danse des mains tendues | Room of the Dance of Outstretched Hands | 摆手厅 |
| Voûte basse | Low Vault | 低厅 |
| Lac du vent | Wind Lake | 浴心池 |
| Galerie du vent | Wind Gallery | 风谷 |
| Cascade I | Waterfall No. I | 洞内瀑布1号 |
| Galerie des cascades | Waterfall Gallery | 瀑布大道 |
| Rivière des cascades | Waterfall River | 瀑布小河 |
| Rivière qui serpente entre les blocs | River that meanders between the Blocks | 石中河 |
| Méandre du rhume | Flu Meander | 盲鱼峡 |
| Cascade du rhume | Flu Waterfall | 望天瀑布 |
| Rivière Régis Gremmel | Régis Gremmel River | 子午谷河 |
| Gouffre Super Tong | Super Tong Shaft | 超级竖井 |

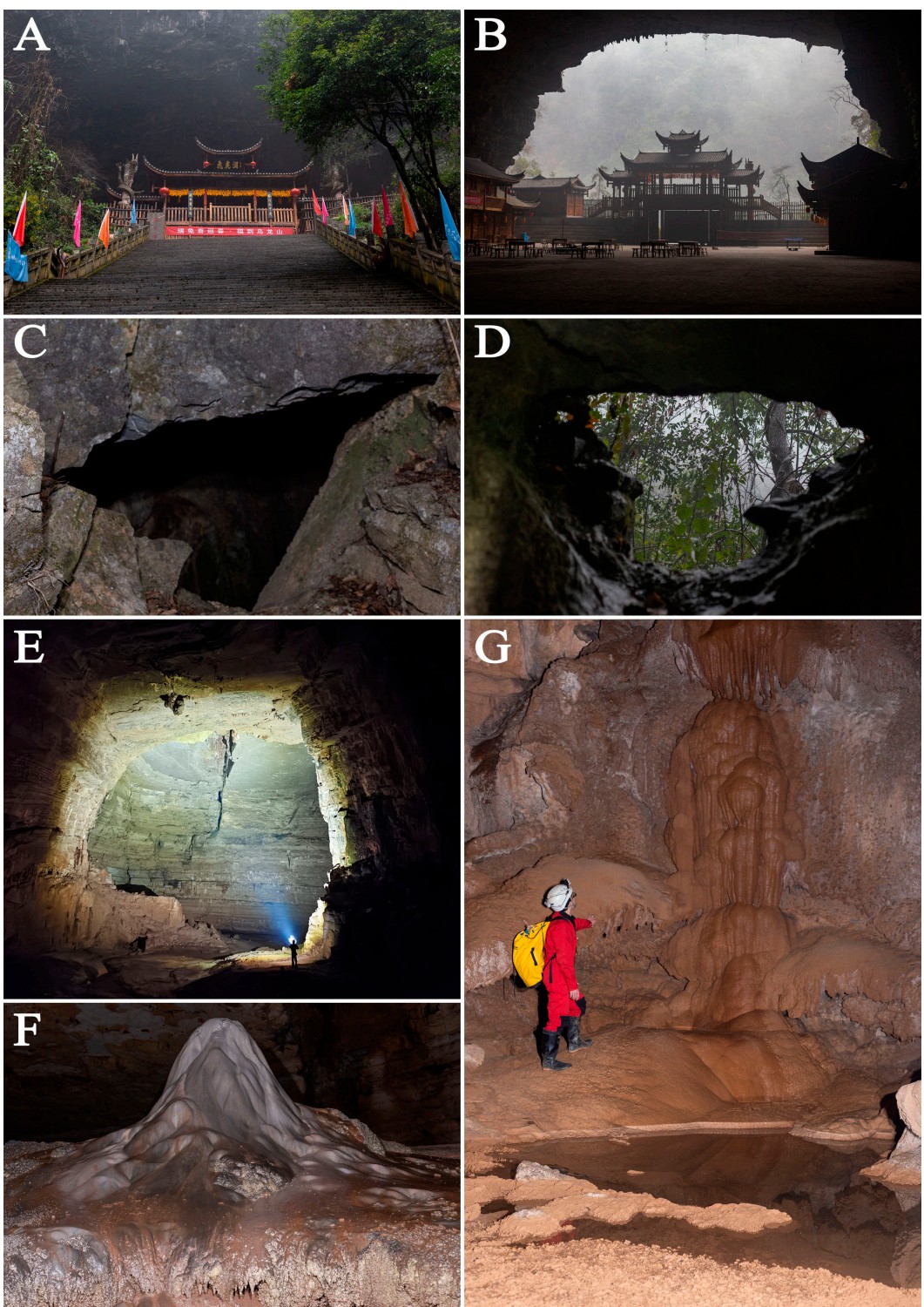

**Figure 4.** Entrances and habitats of Feihu Dong: (**A**,**B**) the main entrance, "Large Porch Entrance"; (**C**,**D**) the entrance of "Tiger Eye"; (**E**) Waterfall Gallery; (**F**,**G**) speleothems.

### 2.2. Sampling

To update the list of cave fauna in Feihu Dong, we reviewed the available literature and conducted explorations from February to April 2023, led by the South China Agricultural University Biocaving Team and Xiangxi Cave Expedition, to upgrade the first biological exploration carried out in August 1995 by Louis Deharveng and Anne Bedos from the Muséum National d'Histoire Naturelle (MNHN Paris) during the Xiangxi 95 expedition. We sampled different habitats within the cave, and we used a combination of direct

sampling, baited traps, and litter extraction methods to collect small invertebrates. Direct sampling was performed by hand or using an aspirator, and baited traps were set up using bananas and fish gut as attractants for terrestrial and aquatic animals. Litter extraction was performed using a sifter to separate soil-dwelling animals from organic matter. All specimens were kept in 75% ethanol for morphological studies and identification or 95% ethanol for DNA sequencing. Photos of the cave animals were taken by a Canon EOS 6D camera (Tokyo, Japan) with a Sigma 50 mm F2.8 EX DG Macro lens (Kanagawa, Japan) and an adapted Meike MK-14 ext E-TTL macro flash (Hongkong, China). They were then processed using Photoshop CC 2019 (San Jose, CA, USA).

*2.3. Terminology*

Ecological classification of cave animals and troglomorphy are defined as in Huang et al., 2021 [18].

**3. Results**

*3.1. Fauna Composition of Feihu Dong*

In total, 62 species of animals have been discovered in Feihu Dong. In the current state of our knowledge, 27 of these species are considered troglobionts (23 species) or stygobionts (four species) taxa, 26 as troglophiles (24 species) or stygophiles (two species), and the remaining nine as trogloxenes (six species) or stygoxenes (three species) (Table 2).

**Table 2.** Species list of cave animals found in cave Feihu Dong. Column status: Tb = troglobiont; Tp = troglophile; Tx = trogloxene; Sb = stygobiont; Sp = stygophile; Sx = stygoxene; * = known endemic of the Huoyan karst or of Feihu Dong; ? = uncertain ecological status or species under study; SCAU = South China Agricultural University.

| No. | Taxon | Taxonomic Classification | Status | Source |
|---|---|---|---|---|
| 01 | cf. *Dugesia japonica* Ichikawa and Kawakatsu, 1964 | Turbellaria: Tricladida: Dugesiidae | Sx ? | [25] |
| 02 | *Angustopila huoyani* Jochum, Slapnik and Páll-Gergely, 2014 | Gastropoda: Stylommatophora: Hypselostomatidae | Tb * | [25,30] |
| 03 | *Synprosphyma* cf. *lyra* (Gredler,1887) | Gastropoda: Stylommatophora: Clausiliidae | Tp | SCAU |
| 04 | Subulinidae sp. | Gastropoda: Stylommatophora: Subulinidae | Tp ? | SCAU |
| 05 | Helixarionoidea sp. | Gastropoda: Stylommatophora: Helixarionoidea | Tp ? | SCAU |
| 06 | Oligochaeta sp. | Oligochaeta | Tx ? | SCAU |
| 07 | *Papiliocoelotes guitangensis* Zhao and Li, 2016 | Arachnida: Araneae: Agelenidae | Tp ?* | [31], SCAU |
| 08 | Agelenidae sp. | Arachnida: Araneae: Agelenidae | Tb | SCAU |
| 09 | Linyphiidae sp. | Arachnida: Araneae: Linyphiidae | Tp ? | SCAU |
| 10 | *Belisana* sp. | Arachnida: Araneae: Pholcidae | Tb | SCAU |
| 11 | Pholcidae sp. | Arachnida: Araneae: Pholcidae | Tb | SCAU |
| 12 | *Telema wunderlichi* Song and Zhu, 1994 | Arachnida: Araneae: Telemidae | Tb * | SCAU |
| 13 | Rhagidiidae sp. | Arachnida: Acari: Rhagidiidae | Tb ? | [25], SCAU |
| 14 | Spinturnicidae sp. | Arachnida: Acari: Spinturnicidae | Tp ? | SCAU |
| 15 | Acari sp. 1 | Arachnida: Acari | Tp ? | SCAU |
| 16 | Acari sp. 2 | Arachnida: Acari | Tp ? | SCAU |
| 17 | Laniatores sp. | Arachnida: Opilionida: Laniatores | Tb | [26] |
| 18 | *Schenkeliobunum* cf. *wuxi* Lu, Wang and Zhang, 2022 | Arachnida: Opilionida: Sclerosomatidae | Tp ? | SCAU |
| 19 | *Glyphiulus deharvengi* Golovatch, Geoffroy, Mauriès and Van Den Spiegel, 2006 | Diplopoda: Spirostreptida: Cambalopsidae | Tb * | [25,32], SCAU |
| 20 | *Epanerchodus* sp. | Diplopoda: Polydesmida: Polydesmidae | Tb * | SCAU |
| 21 | *Eutrichodesmus sketi* Golovatch, Geoffroy, Mauriès and Van Den Spiegel, 2015 | Diplopoda: Polydesmida: Haplodesmidae | Tb * | [25,33], SCAU |
| 22 | cf. *Lithobius* (*Monotarsobius*) sp. | Chilopoda: Lithobiomorpha: Lithobiidae | Tb | SCAU |
| 23 | Geophilidae sp. | Chilopoda: Geophilomorpha: Geophilidae | Tp | SCAU |
| 24 | *Caridina longshan* Cai and Ng, 2018 | Malacostraca: Decapoda: Atyidae | Sb * | [25,29], SCAU |
| 25 | *Gammarus* sp. | Malacostraca: Amphipoda: Gammaridae | Sb | [25], SCAU |
| 26 | *Trogloniscus* sp. | Malacostraca: Isopoda: Styloniscidae | Tb | SCAU |
| 27 | Lernaeidae sp. | Copepoda: Cyclopoida: Lernaeidae | Sb ? | Zhou J.J. |
| 28 | *Coecobrya* sp. | Collembola: Entomobryomorpha: Entomobryidae | Tb | [25], SCAU |
| 29 | *Tomocerus* sp. | Collembola: Entomobryomorpha: Tomoceridae | Tb | [25], SCAU |
| 30 | *Vitronura* sp. | Collembola: Poduromorpha: Neanuridae | Tx | [25] |

| 31 | Campodeidae sp. | Entognatha: Diplura: Campodeidae | Tb | SCAU |
|----|----|----|----|----|
| 32 | *Toshiaphaenops ovicollis* Ueno, 1999 | Insecta: Coleoptera: Carabidae | Tb * | [23], SCAU |
| 33 | *Huoyanodytes tujiaphilus* Tian and Huang, 2016 | Insecta: Coleoptera: Carabidae | Tb * | SCAU |
| 34 | *Cathaiaphaenops* (*Cathaiaphaenops*) *delprati* Deuve, 1996 | Insecta: Coleoptera: Carabidae | Tb * | [21,25], SCAU |
| 35 | *Sinotroglodytes bedosae* Deuve, 1996 | Insecta: Coleoptera: Carabidae | Tb * | [21,25], SCAU |
| 36 | *Zopherobatrus tianmingyii* Yin and Li, 2015 | Insecta: Coleoptera: Staphylinidae | Tb | SCAU |
| 37 | *Nipponobythus* sp. | Insecta: Coleoptera: Staphylinidae | Tp ? | SCAU |
| 38 | *Quedius feihuensis* Smetana, 1999 | Insecta: Coleoptera: Staphylinidae | Tx ?* | [34], SCAU |
| 39 | *Pseudeurostus hilleri* (Reitter, 1877) | Insecta: Coleoptera: Ptinidae | Tx | SCAU |
| 40 | *Mycetina* sp. | Insecta: Coleoptera: Endomychidae | Tx | SCAU |
| 41 | *Tachycines* (*Gymnaeta*) *omninocaecus* (Gorochov, Rampini and Di Russo, 2006) | Insecta: Orthoptera: Rhaphidophoridae | Tb * | [25,35,36], SCAU |
| 42 | *Eutachycines crenatus* (Gorochov, Di Russo and Rampini, 2006) | Insecta: Orthoptera: Rhaphidophoridae | Tb * | [12,25,35], SCAU |
| 43 | *Tachycines* (*Gymnaeta*) *solidus* (Gorochov, Rampini and Di Russo, 2006) | Insecta: Orthoptera: Rhaphidophoridae | Tp ? | [25,35,36], SCAU |
| 44 | Ischnopsyllidae sp. | Insecta: Siphonaptera: Ischnopsyllidae | Tb ? | SCAU |
| 45 | *Sarasaeschna* sp. | Insecta: Odonata: Aeshnidae | Sp ? | SCAU |
| 46 | Perlidae sp. | Insecta: Plecoptera: Perlidae | Sp ? | SCAU |
| 47 | Trichoptera sp. | Insecta: Trichoptera | Sx ? | SCAU |
| 48 | Triphosa sp. | Insecta: Lepidoptera: Geometridae | Tp | SCAU |
| 49 | Tineidae sp. | Insecta: Lepidoptera: Tineidae | Tp ? | SCAU |
| 50 | Anisolabididae sp. | Insecta: Dermaptera: Anisolabididae | Tp ? | SCAU |
| 51 | Culicidae sp. | Insecta: Diptera: Culicidae | Tp ? | SCAU |
| 52 | Limoniidae sp. | Insecta: Diptera: Limoniidae | Tp ? | SCAU |
| 53 | Psychodinae sp. | Insecta: Diptera: Psychodidae | Tp ? | SCAU |
| 54 | *Oreolalax rhodostigmatus* Hu and Fei, 1979 | Amphibia: Anura: Pelobatidae | Tp | [25], SCAU |
| 55 | *Rana* sp. | Amphibia: Anura: Ranidae | Tx ? | SCAU |
| 56 | *Triplophysa xiangxiensis* (Yang, Yuan and Liao, 1986) | Actinopterygii: Cypriniformes: Nemacheilidae | Sb * | [25,28], SCAU |
| 57 | *Misgurnus anguillicaudatus* (Cantor, 1842) | Actinopterygii: Cypriniformes: Cobitidae | Sx ? | SCAU |
| 58 | *Myotis chinensis* (Tomes, 1857) | Mammalia: Chiroptera: Vespertilionidae | Tp | SCAU |
| 59 | *Myotis altarium* Thomas,1911 | Mammalia: Chiroptera: Vespertilionidae | Tp | [37], SCAU |
| 60 | *Rhinolophus pearsonii* Horsfield, 1851 | Mammalia: Chiroptera: Rhinolophidae | Tp | SCAU |
| 61 | *Rhinolophus pusillus* Temminck, 1834 | Mammalia: Chiroptera: Rhinolophidae | Tp | SCAU |
| 62 | *Rhinolophus sinicus* Andersen, 1905 | Mammalia: Chiroptera: Rhinolophidae | Tp | SCAU |

### 3.2. *Notes on Animals Found in Feihu Dong*

Firstly, it has to be stressed that the taxonomic coverage of our species list is biased. The group that contributes the most to aquatic diversity, Microcrustacea, has not been sampled.

Aside from this gap, a total of 62 animal species occurs in Feihu Dong, and Insecta are the most abundant group with 22 recorded representatives. Many of them are classified as troglobionts as their occurrence is limited to cave habitats, and they frequently exhibit troglomorphic characters. The major and interesting groups are listed and discussed below.

#### 3.2.1. Mollusca

Mollusca in Feihu Dong comprise four species. Unidentified Subulinidae (Figure 5A), a very frequent troglophile of tropical caves, and Helixarionoidea (Figure 5B) shells were located inside the Tiger Eye. *Synprosphyma* cf. *lyra* (Figure 5C), found in the Low Vault and Flu Waterfall, appears to be the most common species. This species may enter the cave through the underground water system since the specimens near the waterfall are buried in litter, which arrives from outside. Another minute gastropod, *Angustopila huoyani*, discovered in the entrance corridor [30], was not encountered during the 2023 survey.

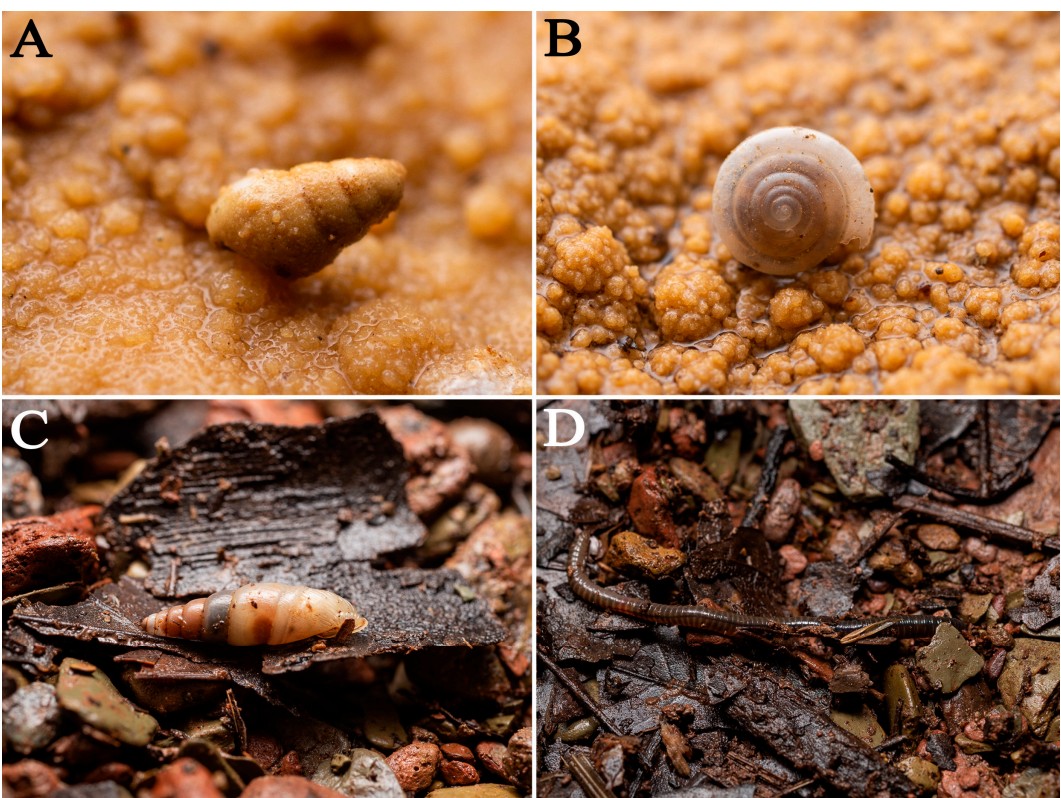

**Figure 5.** Animals found in Feihu Dong: (**A**) Subulinidae sp.; (**B**) Helixarionoidea sp.; (**C**) *Synprosphyma* cf. *lyra* (Gredler,1887); (**D**) Oligochaeta sp.

#### 3.2.2. Arachnida

Six species of spiders are found in various sections of Feihu Dong. *Telema wunderlichi* (Figure 6A), a *Belisana* species (Figure 6B), and an Agelenidae species (Figure 6D) were mostly observed in a moist microhabitat in Tiger Eye, hiding under scattered rubble. A Linyphiidae species (Figure 6C) and *Papiliocoelotes guitangensis* (Figure 6E) were found on the ground of the Low Vault. The unidentified Pholcidae species was collected far inside the cave, after the large chamber; it has extremely long legs and unpigmented, reduced

eyes. We can confidently say that *Telema wunderlichi* is a trogobiont because of its eyelessness. In addition, we assume that *Belisana*, as well as the unidentified Agelenidae, which have reduced pigmentation in the body and eyes, are also trogloblionts. *Papiliocoelotes guitangensis*, though only known from caves, has no adaptive characters related to cave life [31] and is here assumed to be a troglophile.

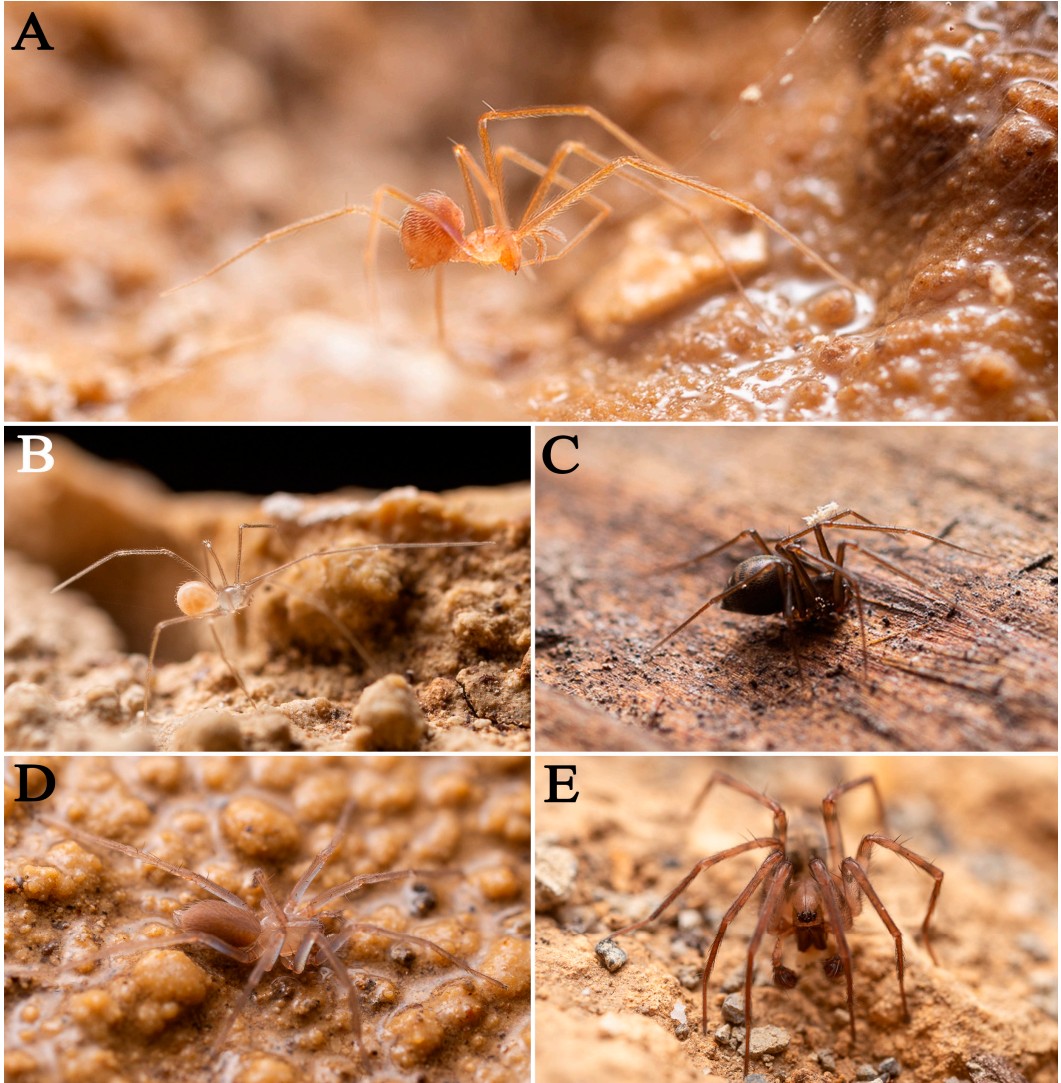

**Figure 6.** Animals found in Feihu Dong: (**A**) *Telema wunderlichi* Song and Zhu, 1994; (**B**) *Belisana* sp.; (**C**) Linyphiidae sp.; (**D**) Agelenidae sp.; (**E**) *Papiliocoelotes guitangensis* Zhao and Li, 2016.

The Opilionida *Schenkeliobunum*, probably *S. wuxi* Lu, Wang and Zhang, 2022 (Figure 7A), discovered in subtropical forest in Chongqing [38], was documented on the ground of Tiger Eye. It is likely a widely distributed species, inhabiting different humid habitats. A single specimen of an unidentified, blind, and highly modified Laniatores was found in the large chamber [26]. It seems to be similar to the cave restricted Opilionida of Southeast Asia, but these usually have eyes.

We collected one species of Rhagidiidae on the ground (Figure 7B). In addition, three ectoparasitic mites, one Spinturnicidae (Figure 7C) and two other unidentified Acari (Figure 7D,E), were found perched on the wings of bats (*Rhinolophus pusillus* Temminck, 1834).

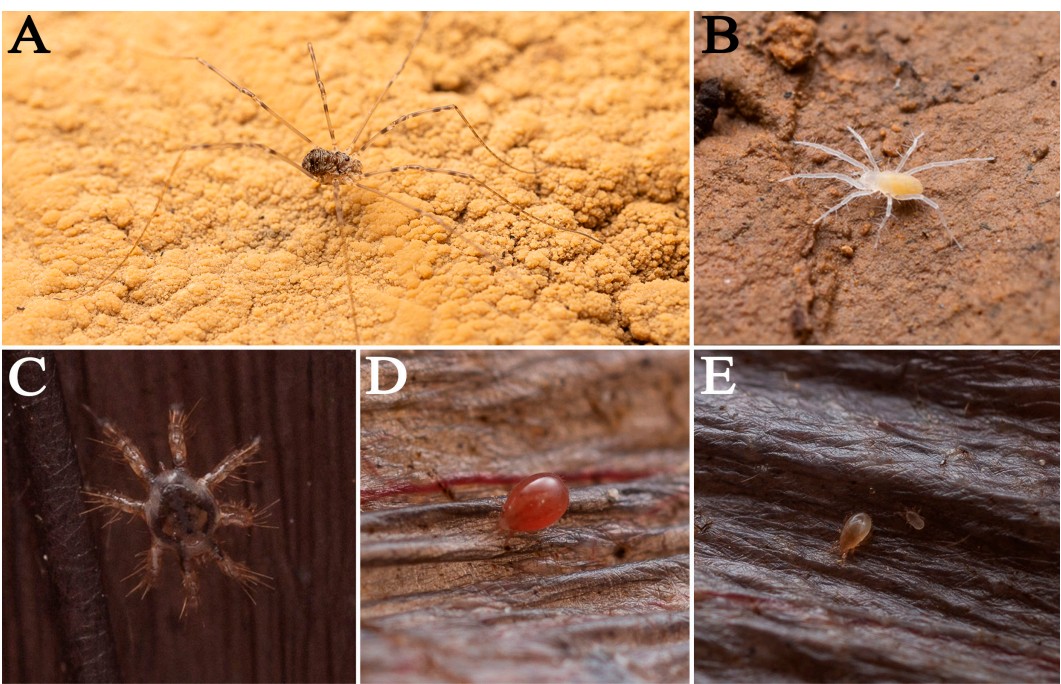

**Figure 7.** Animals found in Feihu Dong: (**A**) *Schenkeliobunum* cf. *wuxi* Lu, Wang and Zhang, 2022; (**B**) Rhagidiidae sp.; (**C**) Spinturnicidae sp.; (**D**) Acari sp. 1; (**E**) Acari sp. 2.

### 3.2.3. Diplopoda

Millipedes are among the most common large invertebrates in Feihu Dong. *Glyphiulus deharvengi* (Figure 8A) and *Eutrichodesmus sketi* (Figure 8B) are widespread in the cave. The presence of these species was expected, as both genera are very frequently found and highly diversified in South China caves [11,14,39]. *Eutrichodesmus sketi* is eyeless. *Glyphiulus deharvengi* has eyes but is unpigmented and is likely a troglobiont. An undescribed *Epanerchodus* species (Figure 8C) has a scattered distribution inside the cave. This *Epanerchodus* exhibits an unusual morphological polymorphism related to its spatial distribution in the cave (Figure 8C,D). However, the different forms recognized were confirmed to be conspecific according to both genital features and preliminary barcoding results.

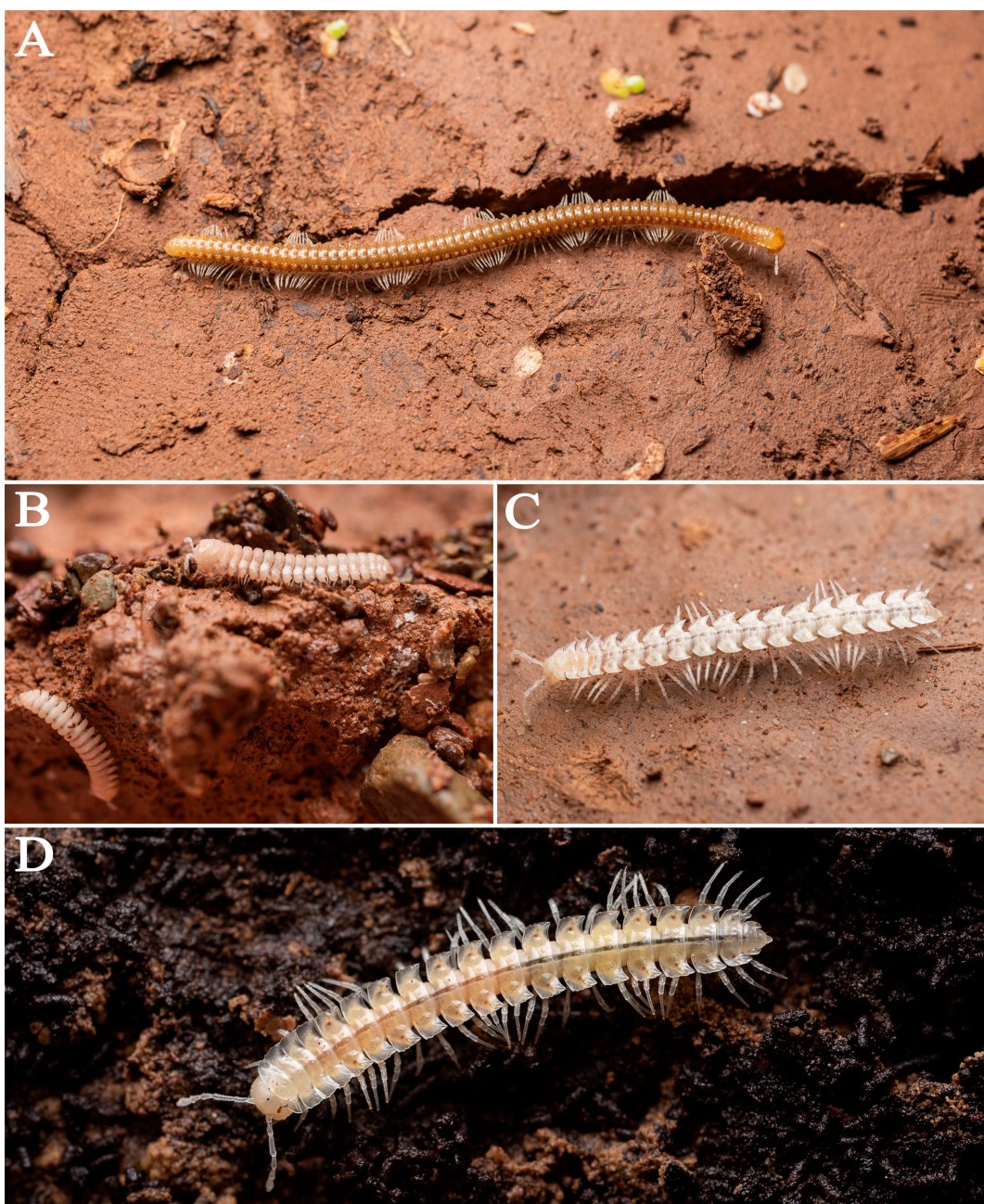

**Figure 8.** Animals found in Feihu Dong: (**A**) *Glyphiulus deharvengi* Golovatch, Geoffroy, Mauriès and Van Den Spiegel, 2007; (**B**) *Eutrichodesmus sketi* Golovatch, Geoffroy, Mauriès and Van Den Spiegel, 2015; (**C**) *Epanerchodus* sp. from Régis Gremmel River; (**D**) *Epanerchodus* sp. from Tiger Eye.

### 3.2.4. Chilopoda

Chilopoda in Feihu Dong comprise two species, viz. *Lithobius* (*Monotarsobius*) sp. (Figure 9A) and a Geophilidae species (Figure 9B). The depigmented *Lithobius* is probably a troglobiont and was found near the Tiger Eye and at a deeper site, Flu Waterfall. In southern China caves, Lithobiidae are rare, with only two *Australobius* species reported from Guizhou and Guangxi [19,40]. The Geophilidae species is probably a troglophile, found only near the Tiger Eye.

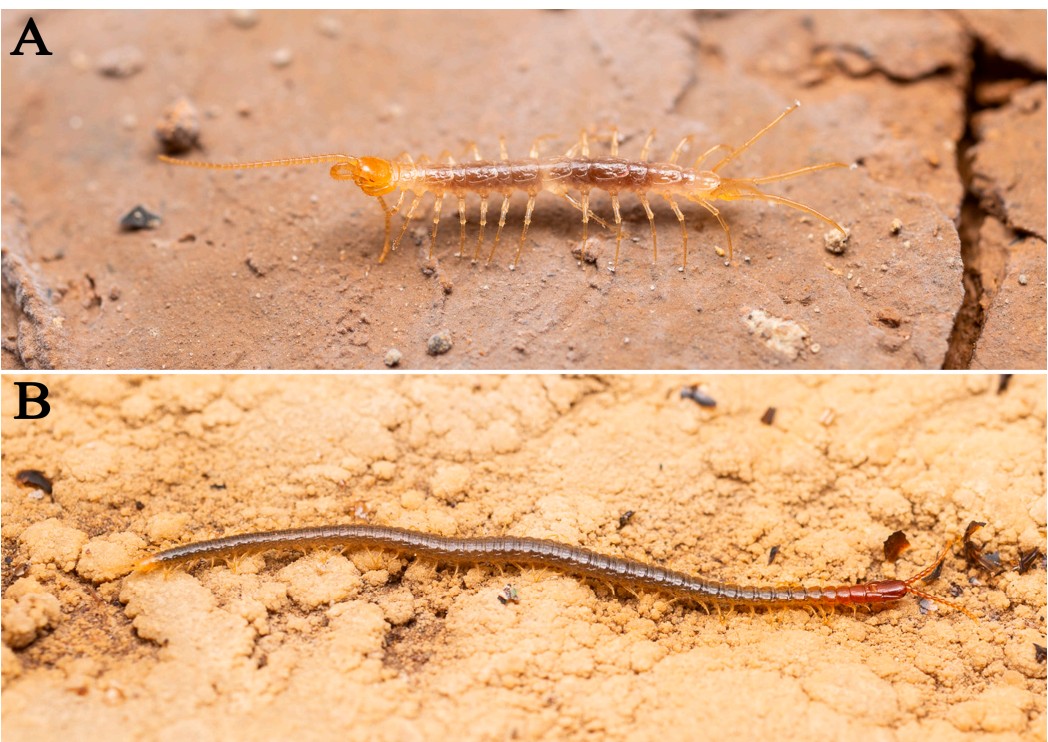

**Figure 9.** Animals found in Feihu Dong: (**A**) *Lithobius* (*Monotarsobius*) sp.; (**B**) Geophilidae sp.

3.2.5. Crustacea

Crustaceans are represented by three moderately troglomorphic species, viz., *Caridina longshan* (Figure 10A), *Gammarus* sp. (Figure 10B), and *Trogloniscus* sp. (Figure 10C). All of them lack either pigment or eyes. *Gammarus* are common in most of the aquatic microhabitats inside Feihu Dong. *Caridina longshan* was collected in Flu Meander, and *Trogloniscus* sp. was collected in the Régis Gremmel River (Figure 4F).

In addition, a species of lernaeid copepod had been observed on the blind fish *Triplophysa xiangxiensis* (Zhou J.J., pers. comm.).

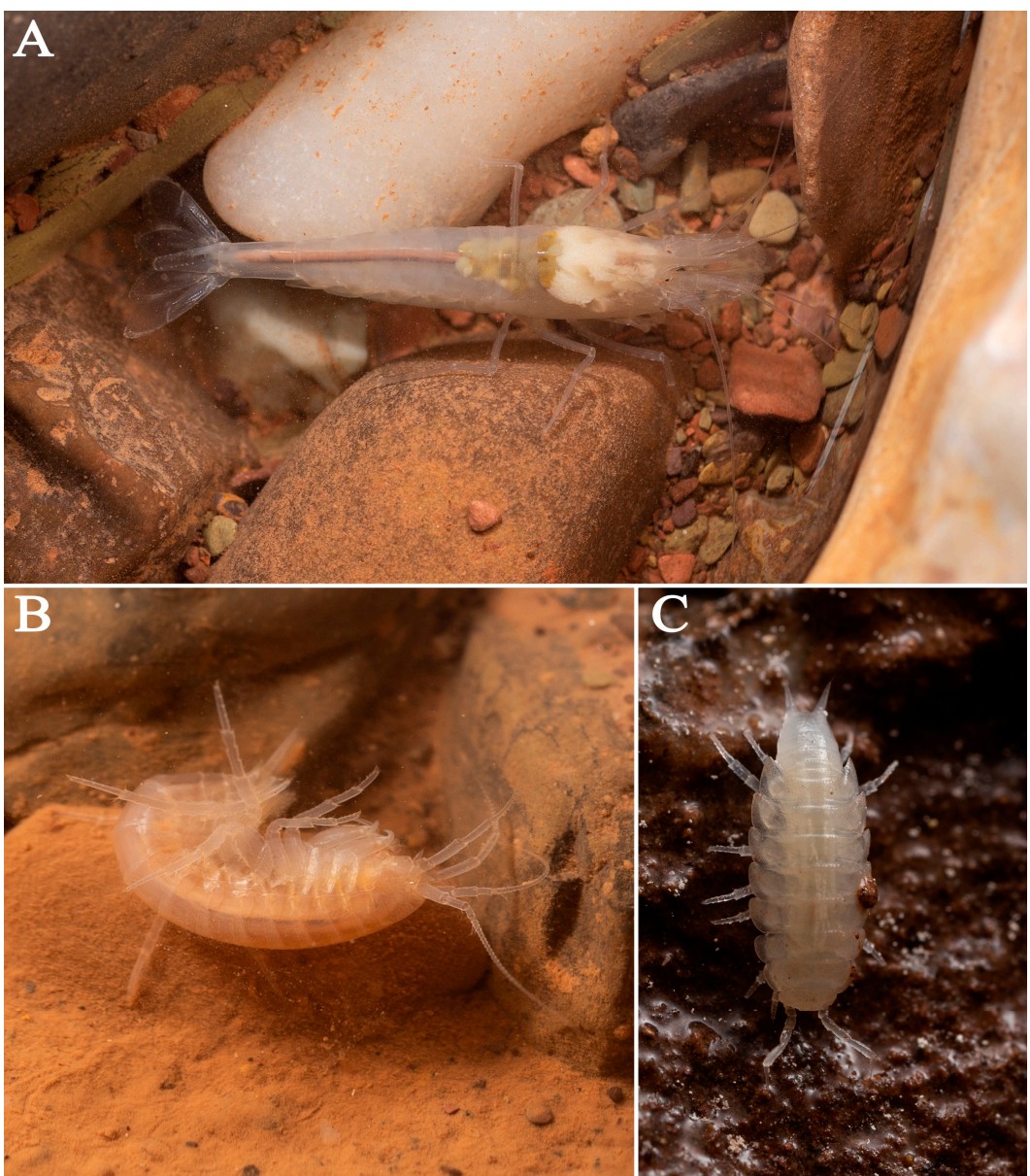

**Figure 10.** Animals found in Feihu Dong: (**A**) *Caridina longshan* Cai and Ng, 2018; (**B**) *Gammarus* sp.; (**C**) *Trogloniscus* sp.

### 3.2.6. Non-Insect Hexapods

A Campodeidae species (Figure 11A) inhabits the dried and sandy habitats of the Régis Gremmel River. It occurs individually and keeps crawling all the time. The combination of its long antennae, cerci, and large size suggests that the species might be a troglobiont. The knowledge of the eight species of Chinese cave-dwelling Campodeidae was summarized by Sendra et al. in 2021 [41]. Remarkable troglomorphic features are obvious in most species, and the Feihu Dong Campodeidae is another highly troglomorphic species of southern China.

Springtails were found on decayed wood and leaves that were brought into the cave by the water flow or human activity. We found species of *Tomocerus* (Figure 11B,C) and *Coecobrya* (Figure 11D) in several collecting points that were associated with predators, e.g., ground beetles and pselaphine beetles. These two springtail genera are highly diversified in southern China's caves.

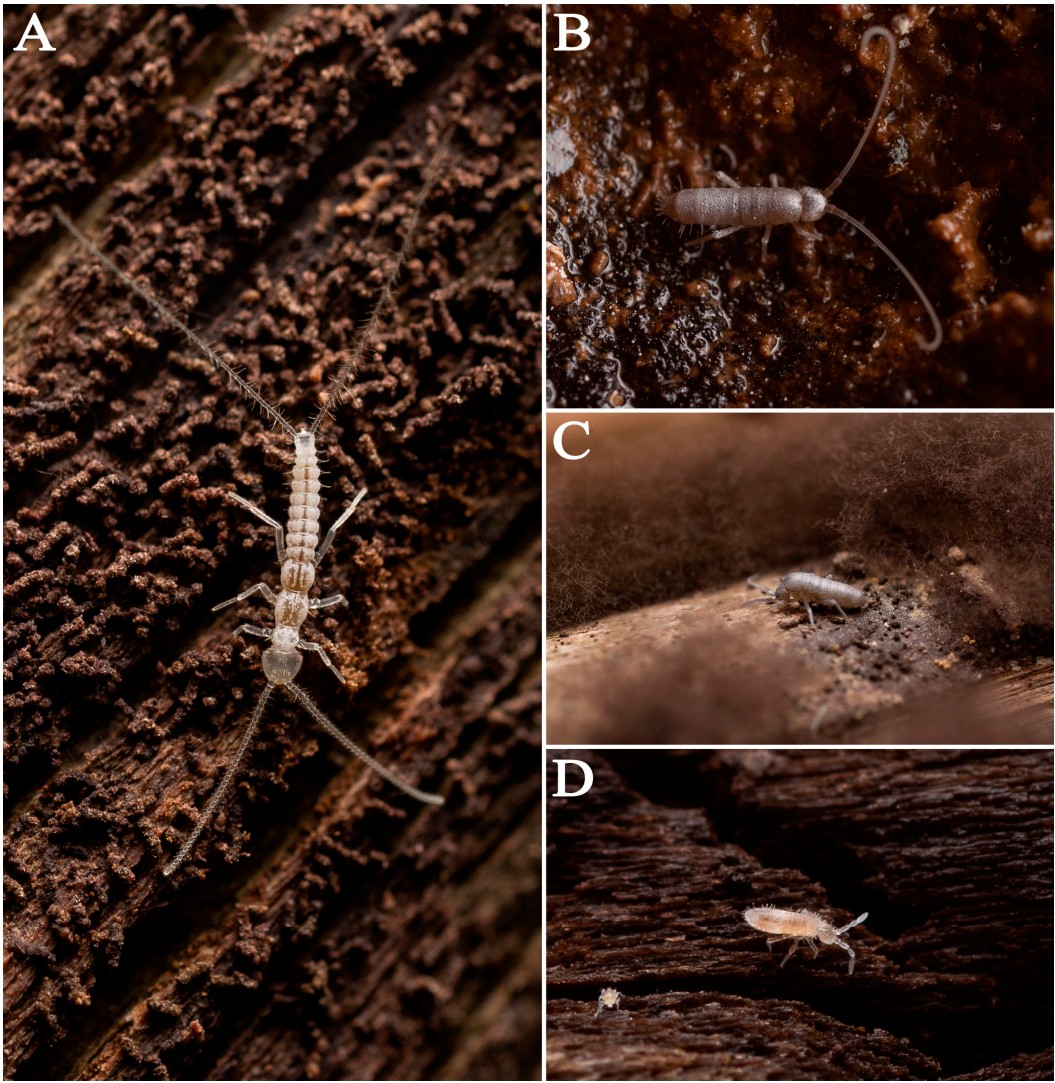

**Figure 11.** Animals found in Feihu Dong: (**A**) Campodeidae sp.; (**B,C**) *Tomocerus* sp.; (**D**) *Coecobrya* sp.

### 3.2.7. Insecta

**Coleoptera**

Nine species of beetles were collected in Feihu Dong, including four Carabidae, three Staphylinidae, one Ptinidae, and one Endomychidae. The troglomorphic carabids *Toshiaphaenops ovicollis* (Figure 12A), *Huoyanodytes tujiaphilus* (Figure 12B), *Cathaiaphaenops* (*Cathaiaphaenops*) *delprati* (Figure 12C), and *Sinotroglodytes bedosae* are widespread in the cave system. During the survey in 2023, the former three species were spotted. In addition, a larva of *C. delprati* (Figure 12D) was captured among an adult population located in Flu Waterfall. Three elytra of ground beetles were uncovered under the compacted sand on the Régis Gremmel River during our survey, which can be attributed to *C. delprati* (two pieces) (Figure 12E) and *T. ovicollis* (one piece) (Figure 12F), respectively. We assume that blind ground beetles are abundant during the rainy season, when strong water flow is carrying lots of resources inside the cave.

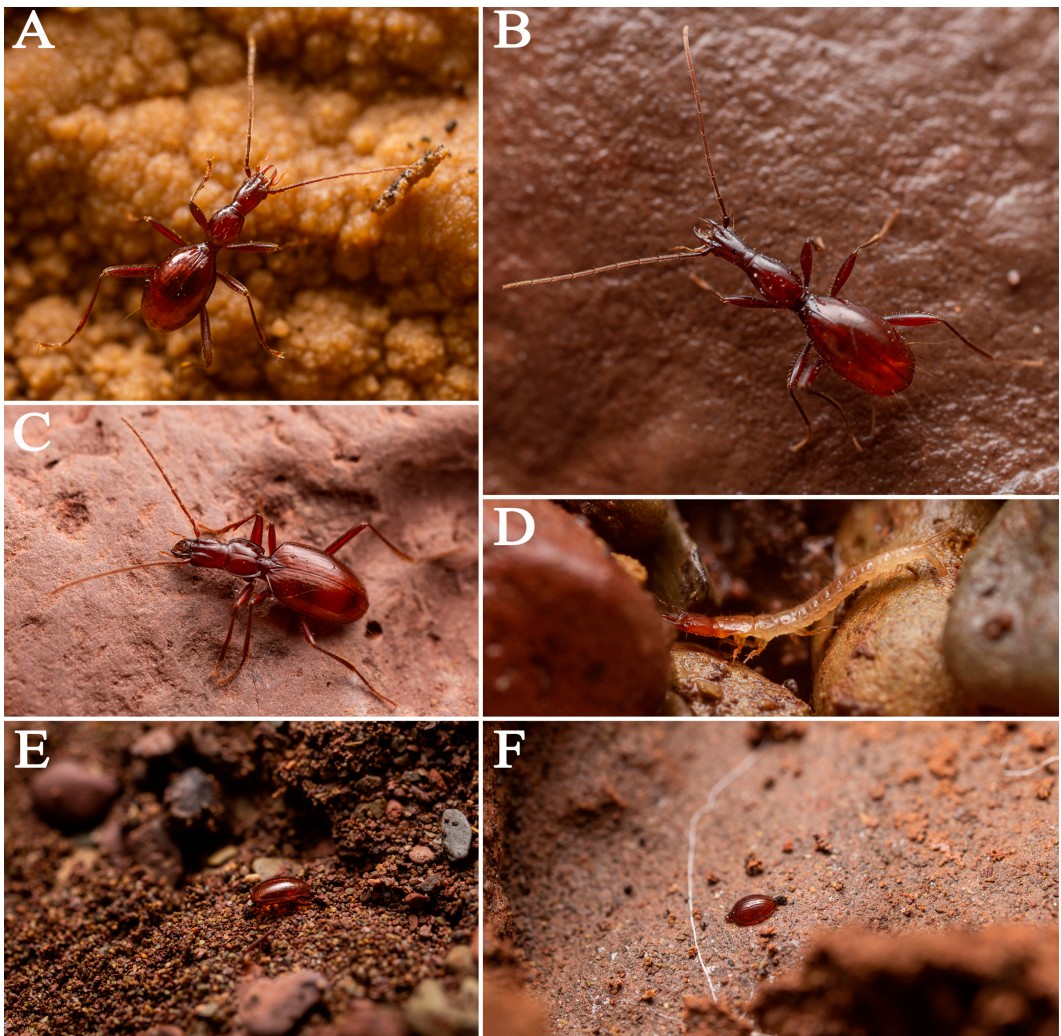

**Figure 12.** Animals found in Feihu Dong: (**A**) *Toshiaphaenops ovicollis* Ueno, 1999; (**B**) *Huoyanodytes tujiaphilus* Tian and Huang, 2016; (**C–E**) adult, larva, and elytra of *Cathaiaphaenops* (*Cathaiaphaenops*) *delprati* Deuve, 1996; (**F**) elytra of *Toshiaphaenops ovicollis* Ueno, 1999.

For staphylinids, the troglomorphic *Zopherobatrus tianmingyii* (Figure 13A), which is also reported from a limestone cave in Guizhou [42], was found near Tiger Eye, accompanied with a lot of springtails, while a possibly troglophile *Nipponobythus* (Figure 13B) species was found alone near the Waterfall No. I. The genus *Zopherobatrus* contains three species, all troglobionts with reduced eyes, previously known from Guizhou, Chongqing, and Sichuan [19]. *Quedius feihuensis* Smetana, 1999 was collected together with *Cathaiaphaenops*. In spite of being only known from Feihu Dong, the species has no troglomorphic character and is likely a trogloxene [34].

In addition, two other beetles are trogloxenes, *viz. Pseudeurostus hilleri* (Figure 13C) and a *Mycetina* species (Figure 13D). They were attracted by baits (banana and fish gut), and we settled in Low Vault.

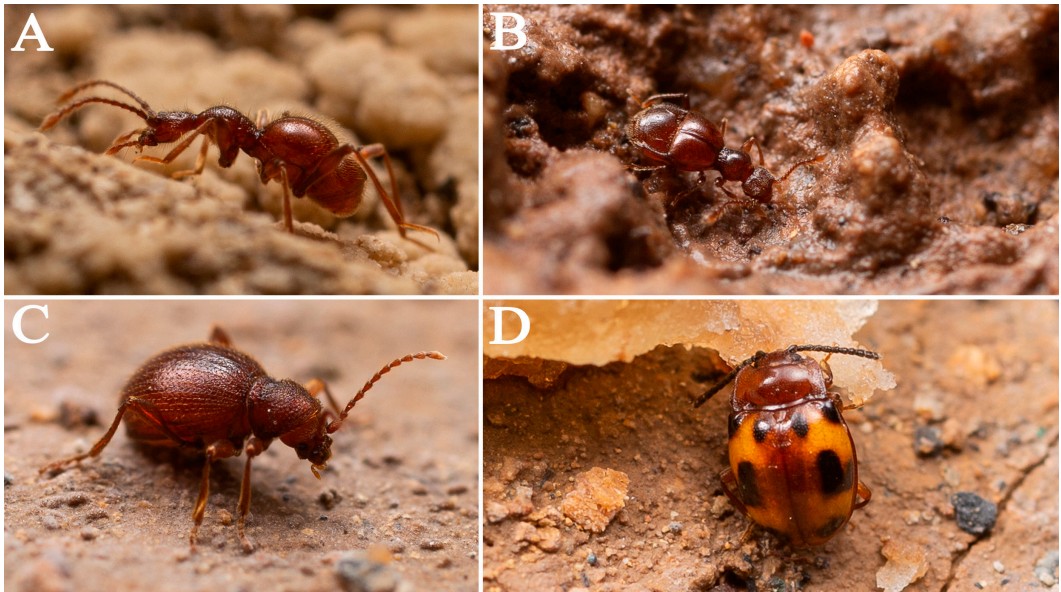

**Figure 13.** Animals found in Feihu Dong: (**A**) *Zopherobatrus tianmingyii* Yin and Li, 2015; (**B**) *Nipponobythus* sp.; (**C**) *Pseudeurostus hilleri* (Reitter, 1877); (**D**) *Mycetina* sp.

**Orthoptera**

Three species of the family Rhaphidophoridae have been spotted in Feihu Dong. *Tachycines* (*Gymnaeta*) *solidus* (Figure 14B), restricted to the Room of the Dance of Outstretched Hands, is probably a troglophile due to its normal-sized eyes and dark, striped body. *Eutachycines crenatus* (Figure 14C) occupies a deeper section of the cave, from Tiger Eye to Wind Gallery, including Low Vault. It has a depigmented body, medium-sized eyes, and a strongly crenate abdomen, which demonstrate its link to cavernicolous life. The third species, *Tachycines* (*Gymnaeta*) *omninocaecus* (Figure 14A), was found along the Régis Gremmel River, far distant from the two aforementioned species. *Tachycines omninocaecus* is a highly troglomorphic cricket on account of its eyelessness and pale body. The degree of troglomorphy seems related to the spatial distribution of this species within the Feihu Dong system when compared to that of the two other cave crickets.

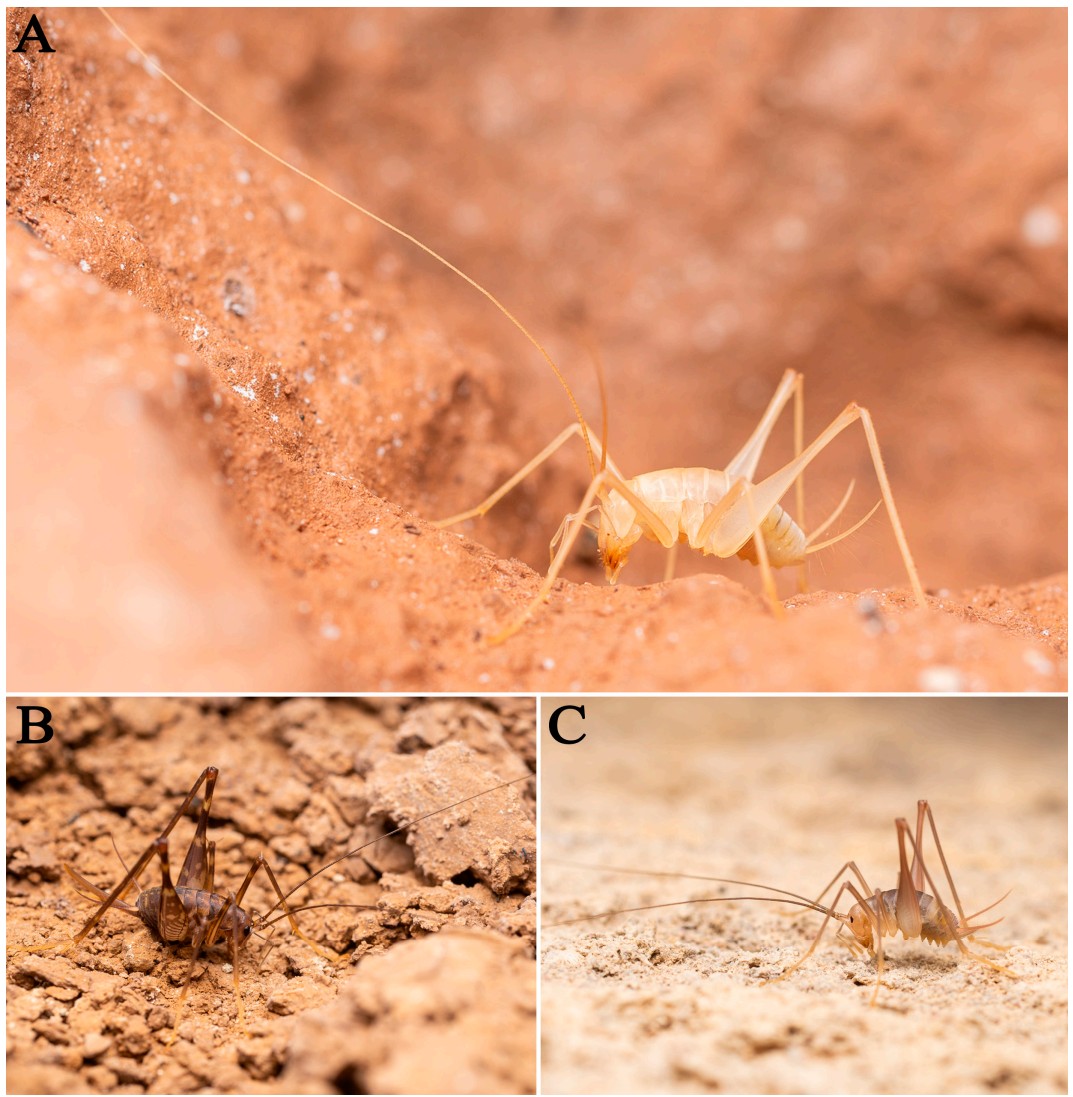

**Figure 14.** Animals found in Feihu Dong: (**A**) *Tachycines* (*Gymnaeta*) *omninocaecus* (Gorochov, Rampini and Di Russo, 2006); (**B**) *Tachycines* (*Gymnaeta*) *solidus* (Gorochov, Rampini and Di Russo, 2006); (**C**) *Eutachycines crenatus* (Gorochov, Rampini and Di Russo, 2006).

**Siphonaptera**

A peculiar flea (Figure 15A), which is non-jumpable and belongs to the family Ischnopsyllidae, was observed by the second author (Zhao M.Z.) (Figure 15B). The species has a rather elongated body and legs. Based on our observations, it seems it crawls slowly and uses its forelegs to detect the environment. Congeneric specimens were also obtained from caves in Guizhou Province and convincingly support their parasitism on bats. Further studies regarding its taxonomy and biology are currently being conducted.

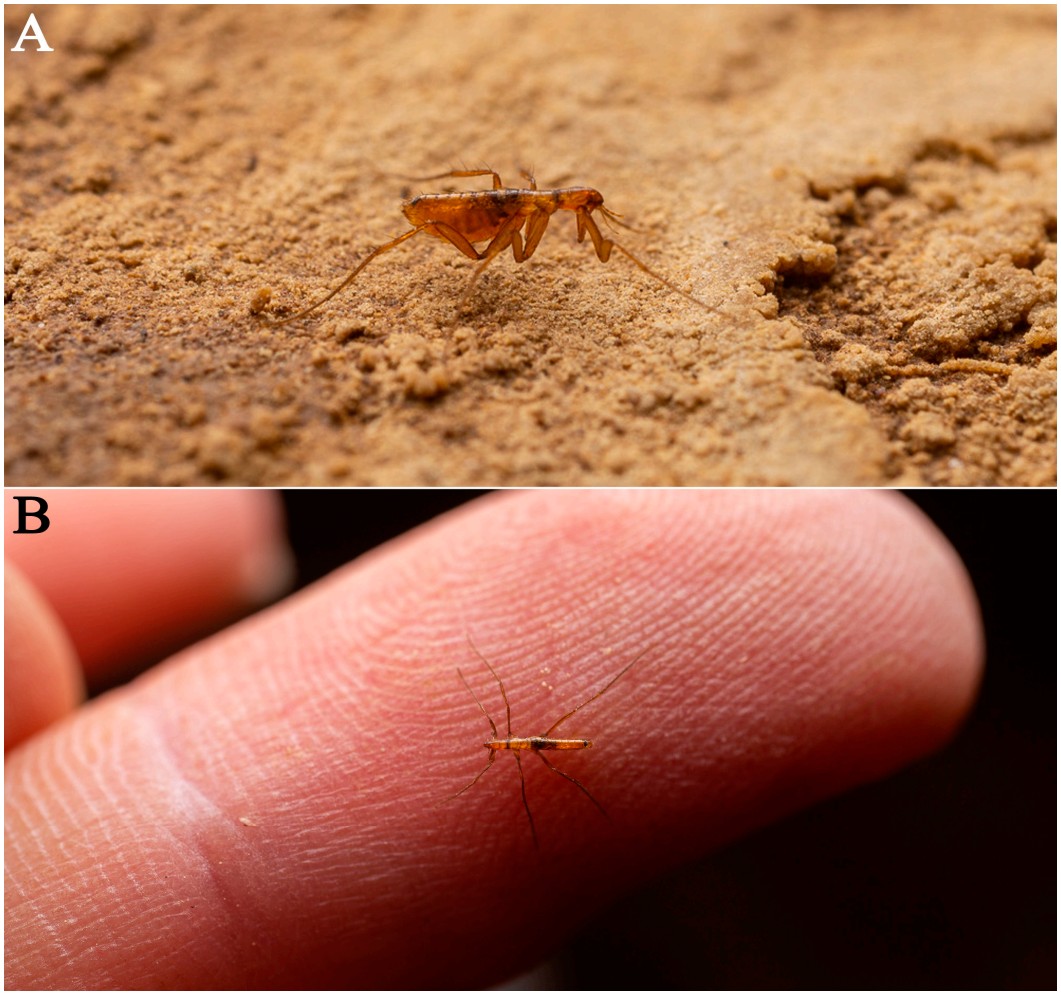

**Figure 15.** Animals found in Feihu Dong: (**A,B**) Ischnopsyllidae sp.

**Aquatic insects**

The aquatic insect fauna in Feihu Dong is represented by three species belonging to Odonata, Plecoptera, and Trichoptera. A *Sarasaeschna* species of the dragonfly family Aeshnidae was found in the Waterfall River. Four final instar (Figure 16A) and one probably penultimate instar (Figure 16B) nymphs were found in the shallow water. Unlike the *Sarasaeschna* species in Ganxiao Cave [18], troglomorphic characters are absent in the final instar nymph of the species found in Feihu Dong. However, we noticed that the younger nymph has a depigmented body and partially developed eyes. The troglophilic status of this species is under investigation. In addition, two mature Trichoptera larvae (Figure 16C) were hidden under the stones of the Régis Gremmel River. We had also uncovered final stage nymphs (Figure 16D) of a Perlidae (Plecoptera) in shallow water of Flu Meander. The discovery of these aquatic insects inside Feihu Dong reveals the complexity of the subterranean water system. The most likely is that a connection exists between the subterranean water and a sinking stream from the surface that remains to be spotted.

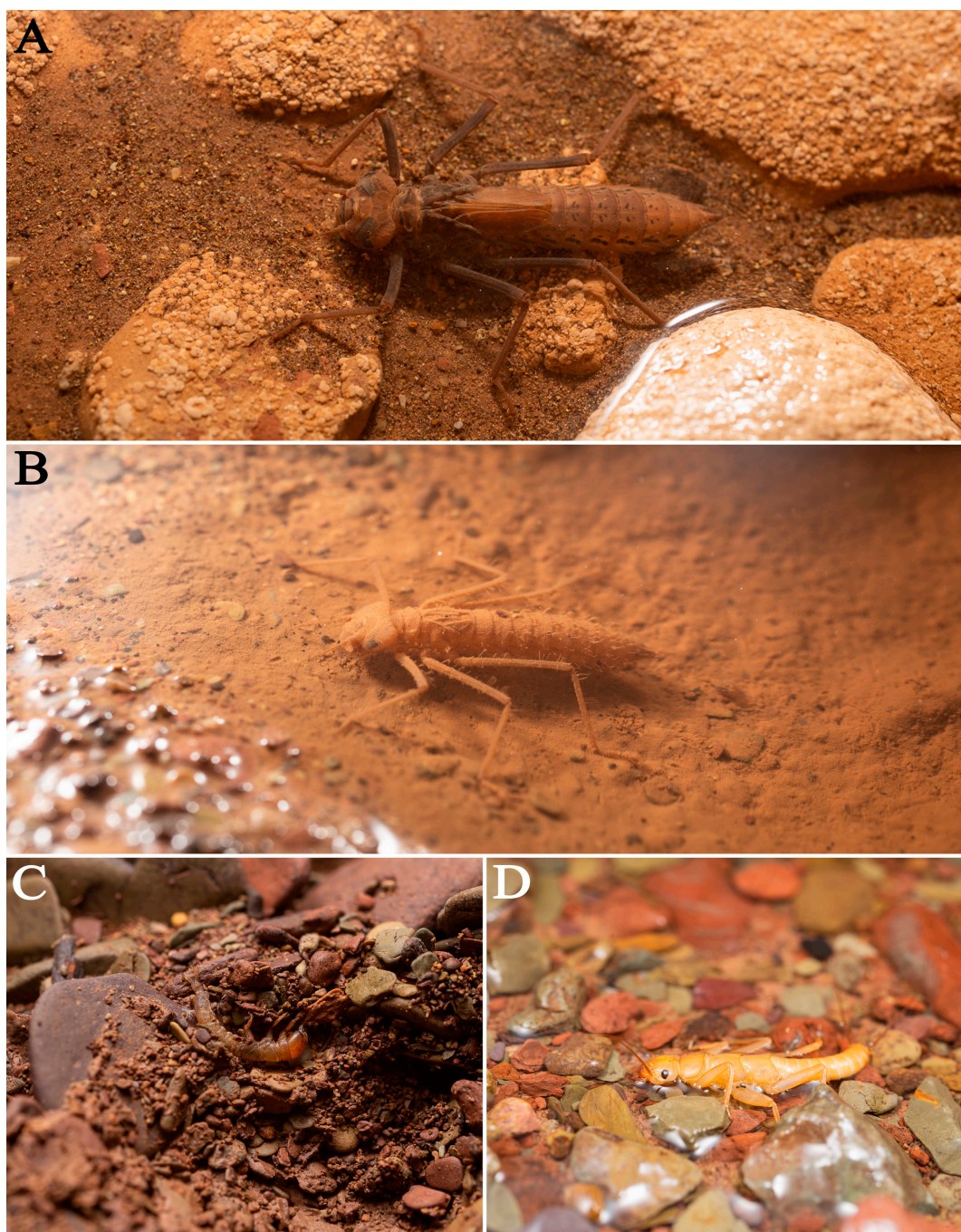

**Figure 16.** Animals found in Feihu Dong: (**A,B**) ultimate and penultimate instar nymphs of *Sarasaeschna* sp.; (**C**) Trichoptera sp.; (**D**) Perlidae sp.

**Other insects**

Triphosa species (Figure 17A), one of the common troglophilic moths in China, were frequently seen on the walls from the entrance to deep sections (*ca.* 0–4 km inside the cave Feihu Dong), sometimes infected by fungi (Figure 17B). Additionally, Tineidae (Lepidoptera) (Figure 17C) and Anisolabididae (Dermaptera) (Figure 17G), as well as Psychodidae (Figure 17D), Culicidae (Figure 17E) and Limoniidae (Figure 17F) (all Diptera), are each represented by one species. Except for the Limoniidae, others are spotted as single individuals. These five species are difficult to assign to an ecological category. However, tineid and psychodid are very often linked to guano in caves, while the three other Diptera belong to families that are among the dominant troglophiles of temperate cave entrances.

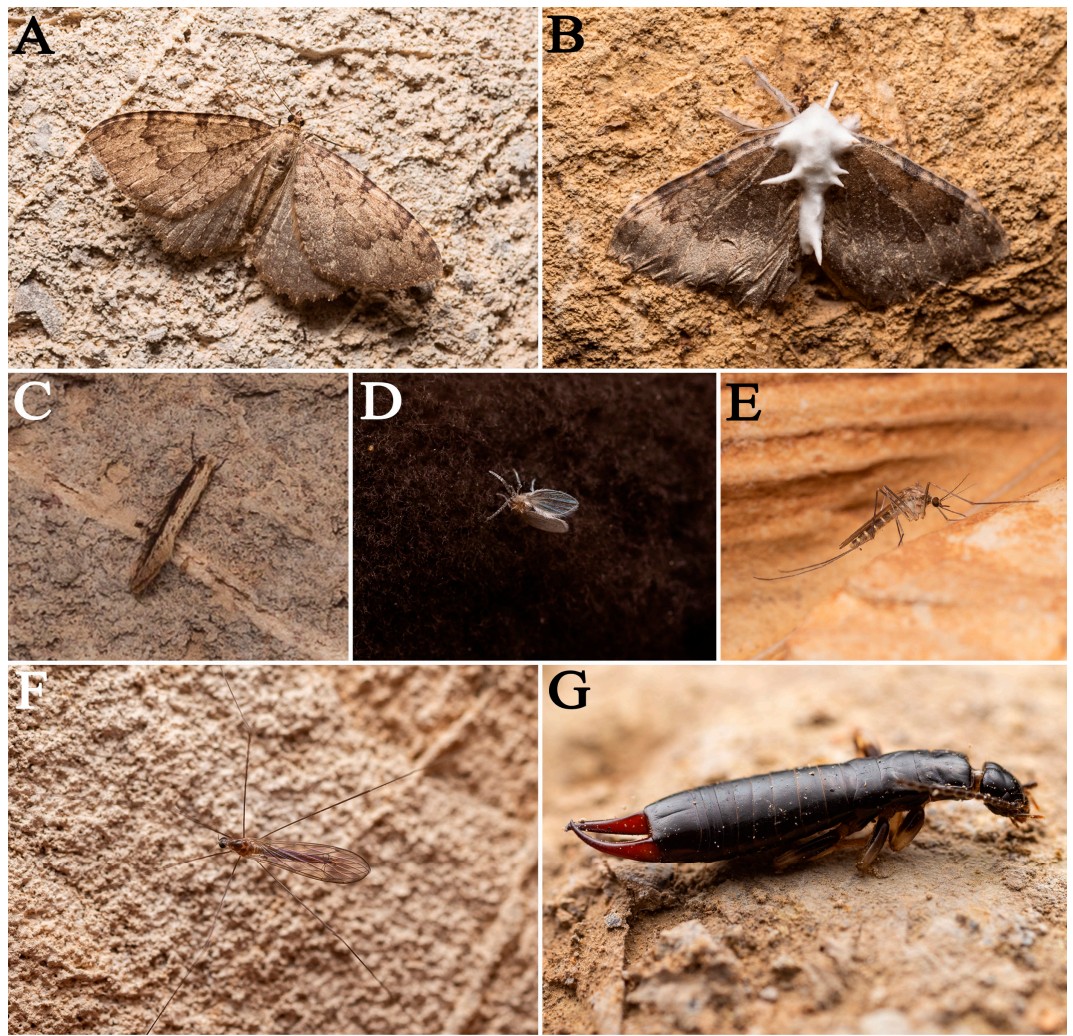

**Figure 17.** Animals found in Feihu Dong: (**A,B**) Triphosa sp.; (**C**) Tineidae sp.; (**D**) Psychodinae sp.; (**E**) Culicidae sp.; (**F**) Limoniidae sp.; (**G**) Anisolabididae sp.

### 3.2.8. Vertebrates

Vertebrates encompass two fish, two frogs, and five bats. The only stygobiotic vertebrate is *Triplophysa xiangxiensis* (Figure 18A), a completely blind fish. The genus comprises 102 species in China, and all the identified cave-dwelling species of the genus were reported from China. More than one fourth of them are typical cavefish, with eyes and pigmentation reduced or completely lost. They are restricted to the karst regions of Yunnan, Hunan, Guizhou, Guangxi, and Chongqing [43–45].

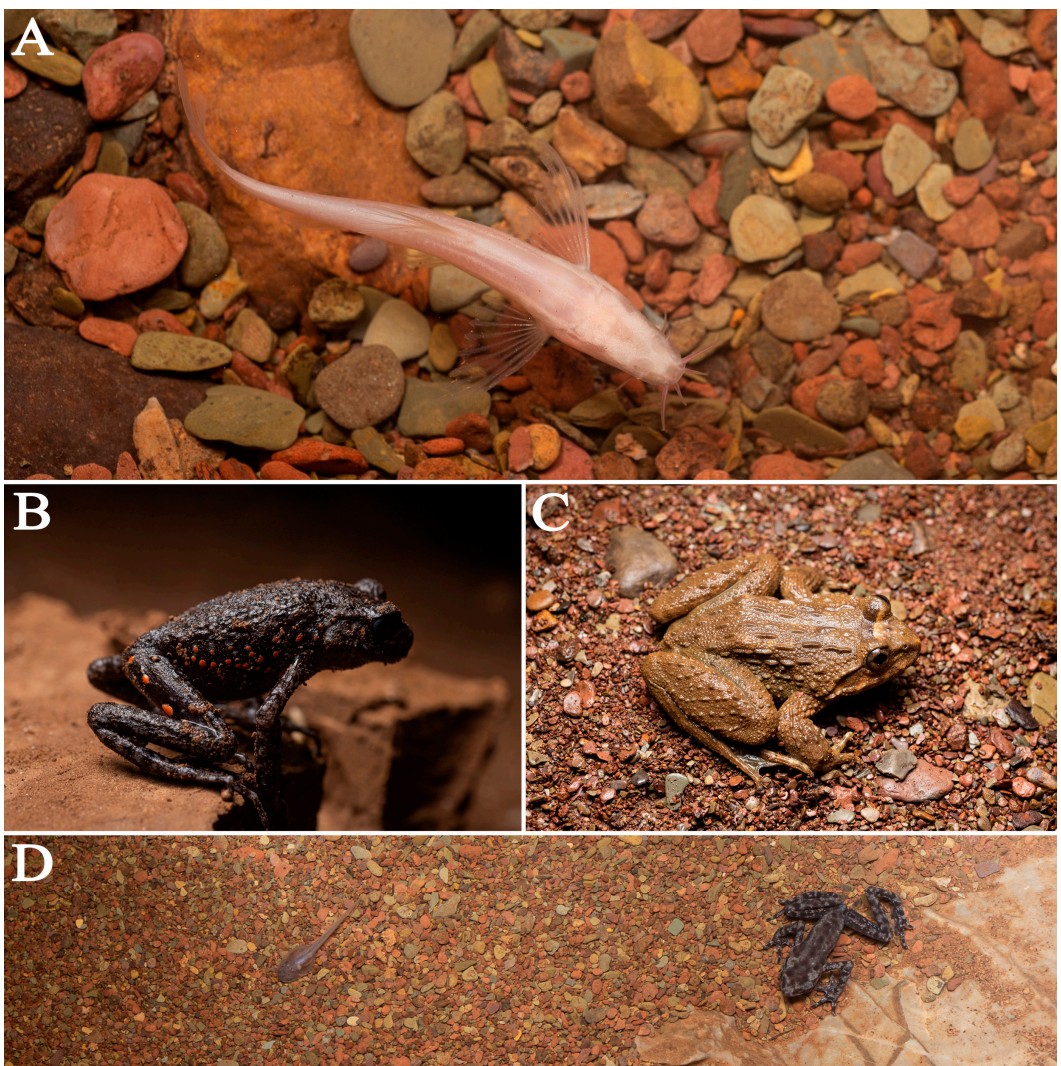

**Figure 18.** Animals found in Feihu Dong: (**A**) *Triplophysa xiangxiensis* (Yang, Yuan and Liao, 1986); (**B**) adult *Oreolalax rhodostigmatus* Hu and Fei, 1979; (**C**) *Rana* sp.; (**D**) larva and adult *Oreolalax rhodostigmatus* Hu and Fei, 1979.

*Triplophysa xiangxiensis* coexists with tadpoles of *Oreolalax rhodostigmatus* (Figure 18B,D), a troglophilic frog, and with the degenerated eyes of the shrimp *Caridina longshan*. Moreover, two trogloxenes, viz., the frog *Rana* sp. (Figure 18C) and the fish *Misgurnus anguillicaudatus,* were discovered accidentally. During winter, bats are scarcely seen along our explored sections inside the cave, yet they accommodate five species (three *Rhinolophus* and two *Myotis*) (Figure 19). Due to the roof of the cave being quite high in most sections, we may have missed some hibernating bats. The bats we observed do not form colonies during the winter season. Their guano is rather scattered in the cave, providing an important source of nourishment for cave-dwelling invertebrates.

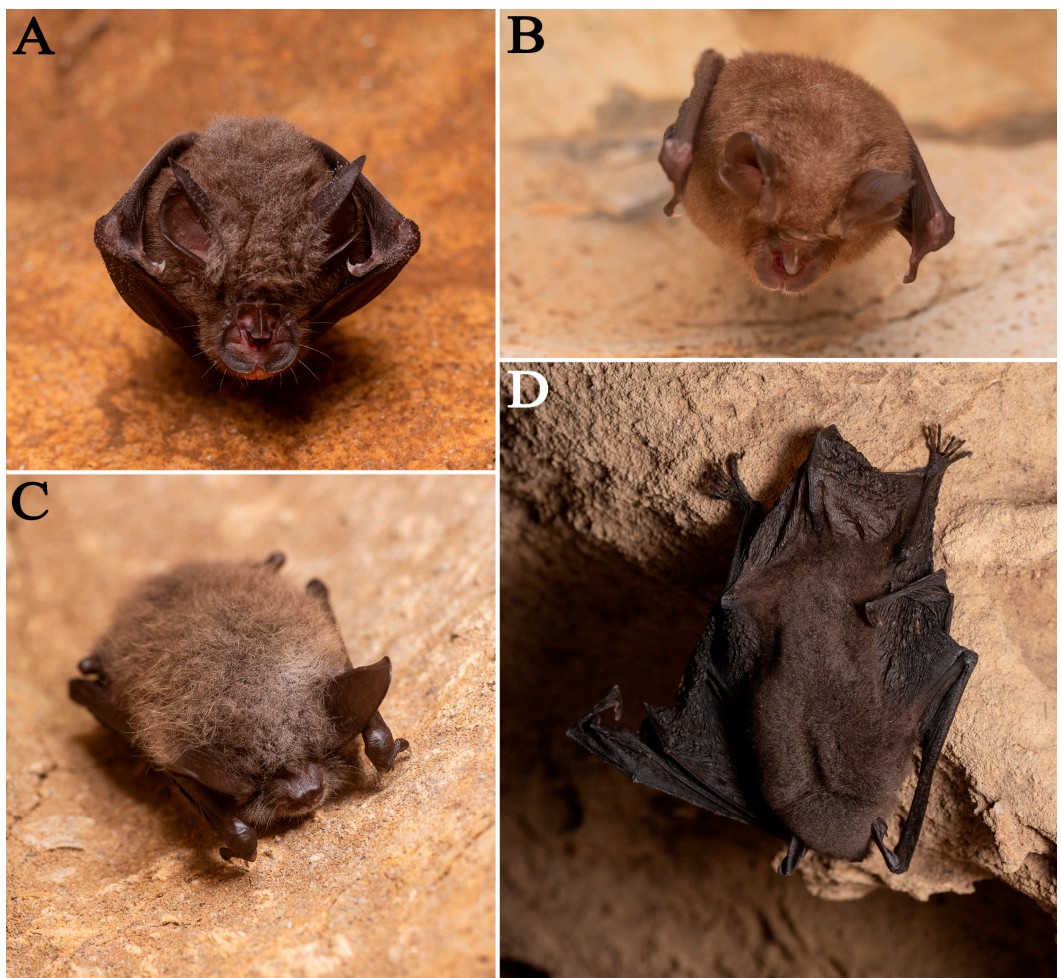

**Figure 19.** Animals found in Feihu Dong: (**A**) *Rhinolophus pearsonii* Horsfield, 1851; (**B**) *Rhinolophus pusillus* Temminck, 1834; (**C**) *Myotis altarium* Thomas, 1911; (**D**) *Myotis chinensis* (Tomes, 1857).

### 3.2.9. Other Animals

Besides what we discussed above, the fauna of Feihu Dong also includes a flatworm and an earthworm. The flatworm may be assigned to *Dugesia japonica*, but the photo is not available [25]. A trogloxene or stygoxene earthworm (Figure 5D) was found in Flu Waterfall, where a lot of organic matter accumulates from outside.

## 4. Discussion

Explorations into the subterranean biodiversity of Feihu Dong revealed an abundance of cave-dwelling fauna, exceeding initial expectations. Here we discuss the implications of these findings for underscoring the significance of Feihu Dong as a site warranting further research and conservation efforts.

### 4.1. Biodiversity Features of Feihu Dong

The subterranean ecosystem of Feihu Dong has been found to be highly diverse and unique, with a total of 27 troglobiotic species recognized in this study. This is a significantly higher number compared to other known caves in South China, such as Ganxiao Dong (20 troglobionts) [18]. Among the different taxonomic groups of subterranean fauna, insects were found to be the most diverse, with eight troglobiotic species identified, including five beetles.

Regarding endemism, 15 of the cited species are restricted to the surrounding karst area. All but one of the 13 troglobiotic (11 species) or stygobiotic (two species) species

identified at the species level are endemic to Feihu Dong or to the karst area of Huoyan, except *Zopherobatrus tianmingyii*, which is also reported from Guizhou [42].

The troglobionts (23 species) and stygobionts (four species) found in Feihu Dong represent a diverse range of taxa that have reached different levels of troglomorphy. Though generally less morphologically modified than species of cave communities in southern Guizhou and northern Guangxi, some species of Feihu Dong are highly troglomorphic, such as the blind ground beetle *Huoyanodytes tujiaphilus* or the blind fish *Triplophysa xiangxiensis*. Interestingly, several of the large stygobionts *Gammarus* sp., *Caridina longshan*, *Triplophysa xiangxiensis*, and the tadpoles of the frog *Oreolalax rhodostigmatus* may occur in high numbers (such as *G.* sp. in Waterfall River) and may even co-occur (such as the tadpoles of *O. rhodostigmatus*, *T. xiangxiensis*, and *C. longshan* in Flu Meander).

### 4.2. Discussion on Some Animals Found in Cave Feihu Dong

Millipedes are commonly seen in caves, where they typically rest on cave walls or attach themselves to decayed wood. In our 2023 investigations, we did not observe large numbers of millipedes in Feihu Dong. This may be due to a lack of food, as bats are known to hibernate during the winter and water flow is often reduced.

Reduced eyes *Caridina* (with cornea pigmentation ranging from totally absent to a small black spot) and blind *Gammarus* are known by many narrowly distributed species from several caves in south China [46–48], mostly described in recent years. Their presence in Feihu Dong is in line with this distribution. *Trogloniscus* sp. belongs to an oligospecific genus of more or less amphibious species known from Guangxi, Guizhou, and Guangdong [19,49]. Its presence in Feihu Dong extends significantly to the north of the distribution of the genus.

Regarding the blind carabid beetle, *Cimmeritodes* (*Cimmeritodes*) *huangi* Deuve, 1996, reported from the cave Baiyan Dong in Huoyan Karst [21]. It is less than 1 km between Baiyan Dong and Feihu Dong in straight-line distance and the beetle is very likely to occur in Feihu Dong.

Some of the unidentified troglobiotic or stygobiotic species may also be endemic to the area, e.g., the springtails *Coecobrya* sp. and *Tomocerus* sp., the spiders *Belisana* sp. and Agelenidae sp., the Chilopoda *Lithobius* (*Monotarsobius*) sp., the Gammaridae *Gammarus* sp., the woodlice *Trogloniscus* sp., and highlighting the need for further investigation and taxonomic research in this region.

### 4.3. Threats and Conservation

One of the threats to the biodiversity of Feihu Dong is the potential impact of tourism. With the increasing popularity of the cave as a tourist destination, excessive tourism development and overcrowding have the potential to negatively affect the cave's ecosystem. It is important to note that the impact of tourism on the cave biodiversity is not yet significant, as its development has been limited to the entrance area, which is, like in all touristic caves, severely disturbed, while kilometers of very large undisturbed galleries exist beyond this area. However, to prevent future negative impacts on biodiversity, it would be useful to maintain the majority of passages, chambers, and cave floors in their original state. For this purpose, several measures could be taken. Firstly, limiting the extent of touristic passages and the number of visitors permitted in the cave at any time. Secondly, monitoring cave biodiversity and conducting regular scientific research as a background to sound conservation measures. Thirdly, encouraging a moderate development that would preserve most cave passages. Fourthly, keep the cave entrance open and not subject to drastic human impacts, to allow the easy passage of bats in and out of the cave. It is less the problem of the entrance itself, which is very large, than of human activity at this entrance, which should remain reasonable in terms of light and noise. It may be beneficial to consider limiting festivals or infrastructural works within the cave from the main entrance to the "Room of the Dance of Outstretched Hands" to further reduce human impact on the cave's ecosystem. Fifthly, it is crucial to provide adequate protection for the unique

and best documented cave species found in Feihu Dong, the blind fish *Triplophysa xiang-xiensis*, which had been assessed for The IUCN Red List of Threatened Species in 1996 and was listed as Vulnerable under criteria D2 [50]. It is also classified as Category II in the list of National Key Protected Wild Animals in China [51]. Moreover, two species of Carabidae of Feihu Dong (*Cathaiaphaenops delprati*, *Huoyanodytes tujiaphilus*), which are the best known invertebrates of the cave, have been recently assigned to the IUCN category "Data Deficient" [52,53], highlighting the fact that further investigations on the Feihu Dong invertebrate fauna are needed to understand the magnitude and extent of the local cave biodiversity. The current and fast development of tourism and associated potential disturbance in the area should be carefully followed during the coming years, in completing for aquatic microcrustacea the baseline inventory proposed here, in assessing the vulnerability of the other troglobionts of the cave system, and in using some of them to monitor changes in biodiversity. That would provide the background needed to implement appropriate conservation measures if they become necessary.

*4.4. Limitations and Prospects*

The subterranean biodiversity of Feihu Dong is far from being fully explored and understood. Although several surveys have been conducted, there are still limitations and prospects that need to be addressed to have a comprehensive understanding of the subterranean biodiversity of Feihu Dong.

One limitation is that the survey area is still relatively limited, and most deeper areas have yet to be explored. Regarding the sampling effort in the aquatic environment, we conducted very few investigations concerned with the baited traps and litter extraction in the water. The aquatic fauna of the Feihu Dong system remains poorly known, and further surveys are needed to fully understand the diversity and ecology of this unique ecosystem. Another gap to fill is that many species have not been identified yet, making their ecological traits difficult to assess. We have not been able to determine the relationship between some of the trogloxenes or stygoxenes and the cave environment, given our poor knowledge of cave fauna in the region. The group to which these species belong has also sometimes been reported from caves. For example, the staphylinid beetle *Quedius feihuensis* was only found in Feihu Dong and is considered here as a trogloxene based on morphology, but other species of the same genus and of similar morphology are also reported from caves [54,55]. This suggests that it may be benefiting from the cave environment in some way. So, we took into account trogloxenes and stygoxenes in this paper. We also face the challenge of finding taxonomists who can deal with the selected material, which is also a classical issue in countries where cave fauna remains under-investigated. Overcoming these limitations would require more extensive and in-depth surveys across the Feihu Dong system, and in the surrounding karst.

There is therefore a significant potential for further research on the subterranean biodiversity of Feihu Dong. The prospects are still far-reaching, as many small passages and connections have not been explored. Systematic surveys of the cave fauna will be carried out in parallel with the four-year exploration project of Chinese cavers, which is expected to shed more light on the subterranean biodiversity of Feihu Dong.

Moreover, the newly discovered passages of Feihu Dong may also potentially connect to other caves, such as Tujiamei Dong (Chushui Dong or Parking Cave "Grotte du Parking"), which could increase the richness of species in the area. Tujiamei Dong, which has been partly surveyed, is very likely linked to Feihu Dong as an outlet for water (Huang, S.B., pers. comm.). It has some species in common with Feihu Dong: the terrestrial species of four cave ground beetles, *Toshiaphaenops ovicollis*, *Huoyanodytes tujiaphilus*, *Cathaiaphaenops delprati* and *Sinotroglodytes bedosae* [56], and millipede *Glyphiulus deharvengi*; as well as a large number of aquatic species, including shrimps *Caridina longshan*, blind fish *Triplophysa xiangxiensis*, and mud fish *Cobitidae* sp. (Figure 20E). But it has other species that are absent from Feihu Dong: the millipede *Epanerchodus tujiaphilus* Liu and Golovatch, 2018 (Figure 20B) [57], a widespread troglophilic scutiger *Thereuopoda*

*clunifera* (Wood, 1862) (Figure 20A), a spider *Nesticella huomachongensis* Lin, Ballarin and Li, 2016 (Figure 20C), and a nymph of damselfly Synlestidae sp. (Figure 20D).

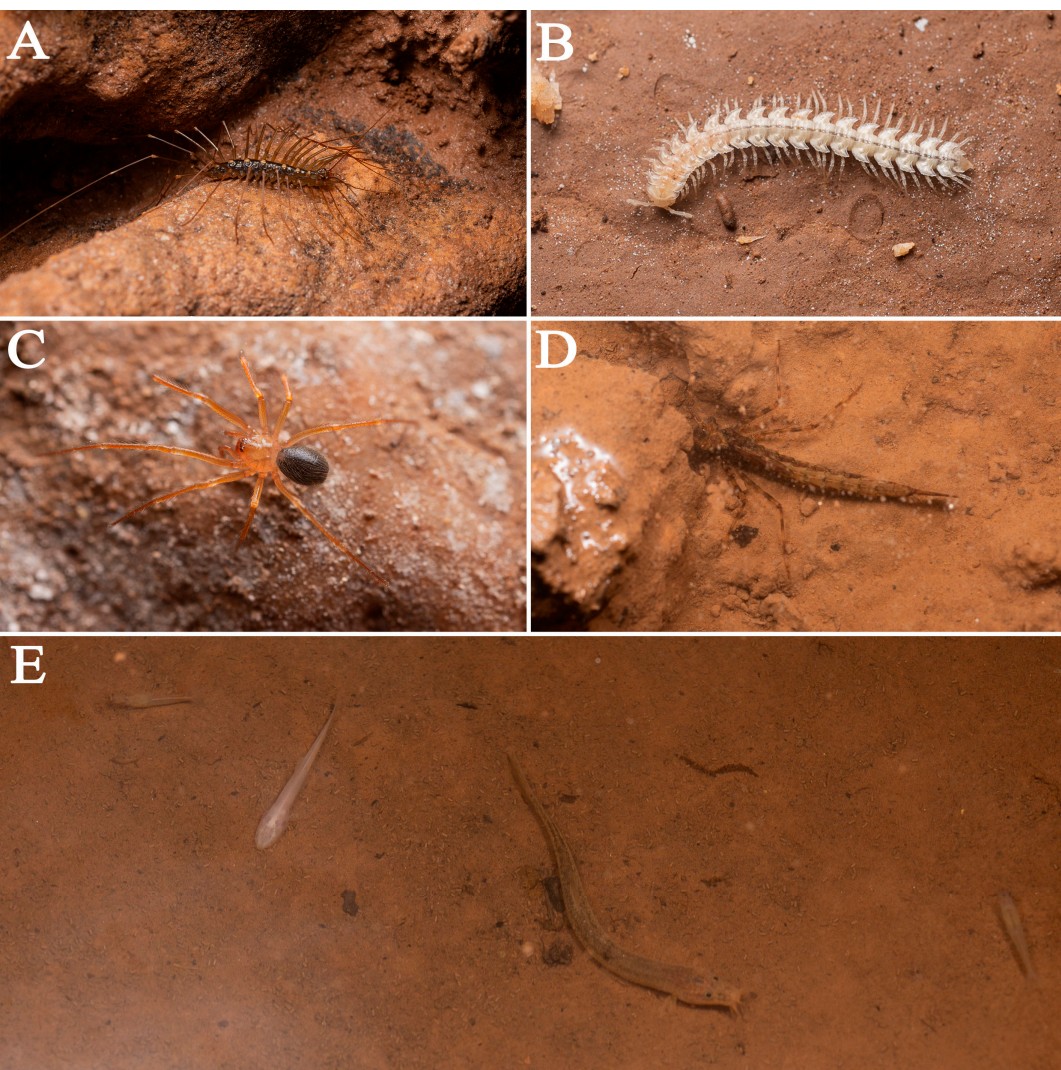

**Figure 20.** Animals found in Tujiamei Dong: (**A**) *Thereuopoda clunifera* (Wood, 1862); (**B**) *Epanerchodus tujiaphilus* Liu and Golovatch, 2018; (**C**) *Nesticella huomachongensis* Lin, Ballarin and Li, 2016; (**D**) Synlestidae sp.; (**E**) shrimps *Caridina longshan* Cai and Ng, 2018, blind fish *Triplophysa xiangxiensis* (Yang, Yuan and Liao, 1986); and mud fish Cobitidae sp.

Moreover, further caves, Da Dong, Panlong Dong (Pa Long Dong), and Mao Dong, might also potentially be connected to Feihu Dong (Figure 1), as well as three potential shafts (Wang Y., pers. comm.) that may act as sources of food through the dripping or sinking waters (Figure 3). The three shafts are positioned above the cave system of Feihu Dong, but their connection with Feihu Dong would require further exploration. Numerous other caves that exist around Huoyan (in Longshan, Rongshun, and Sangzhi regions) have been less intensively, or not at all surveyed biologically. The rare available records that have been published indicate that they probably possess levels of species richness similar to those of Feihu Dong.

Although we know the significance of water for cave fauna, our understanding of the hydrogeology of Feihu Dong is still limited. No validated surface streams were addressed, except for the shaft 'Gouffre Super Tong' and three potential shafts (Figure 1) that may act as inlets for rainwater during the rainy season. The relationship between the cave's water system and the external water system is not well understood and requires further research.

In conclusion, the study of Feihu Dong has exceeded expectations in terms of species richness, and it is likely that many other caves in China have a similar level of subterranean biodiversity. A further significant increase in species richness in South China's caves and karsts can be expected in the near future. Even if research on cave fauna in China still lags behind that of some western countries [58–60], the results obtained on Feihu Dong are demonstrating the potential for South China to become a world hotspot for cave biodiversity. This is indeed stimulating for further investigations, even if there is still a long way to go to fully uncover the diversity of cave fauna in the huge karsts of South China. It is demanding to expand the scope and intensity of cave fauna surveys, and to explore and document the intricate subterranean habitats of caves and karsts in China through collaborative work.

**Author Contributions:** Conceptualization, M.T., W.L. and S.H.; methodology, S.H., M.Z. and W.L.; software, S.H., M.Z. and W.L.; validation, M.T., A.B., X.L., Y.W., M.C. and W.L.; formal analysis, S.H., M.Z. and W.L.; investigation, S.H., M.Z., A.B., M.C. and Y.W.; resources, Y.W., M.C., A.B. and S.H.; data curation, S.H. and M.Z.; writing—original draft preparation, S.H., M.Z., X.L. and W.L.; writing—review and editing, S.H., M.Z., A.B., M.T., X.L., M.C. and W.L.; visualization, S.H. and M.Z.; supervision, M.T. and W.L.; project administration, M.T. and W.L.; funding acquisition, M.T. and W.L. All authors have read and agreed to the published version of the manuscript.

**Funding:** This research received no external funding.

**Institutional Review Board Statement:** Not applicable.

**Data Availability Statement:** Not applicable.

**Acknowledgments:** We would like to express our sincere gratitude to various organizations, individuals, teams who have contributed to the success of our research project on the subterranean biodiversity of Feihu Dong. We extend our appreciation to the Speleology Geological Professional Committee of the Geological Society of China, Xiangxi UNESCO Geopark, Wulongshan National Geopark, and Wulongshan Canyon Scenic Area for their support and cooperation throughout the project. We would also like to thank Yuanhai Zhang of the Institute of Karst Geology, Chinese Academy of Geological Sciences, for arranging the biocaving in Feihu Dong, which was instrumental in our research efforts. Special thanks go to Louis Deharveng from the Muséum National d'Histoire Naturelle for his efforts to inform the foundation about the biodiversity in Feihu Dong, Jian Zhou (member of the team Xiangxi Cave Expedition) for sharing valuable information and data about Feihu Dong, and Zhi Qian of the Guizhou Speleology Association for providing information about Shuanghe Dong. We are also grateful to the following individuals who assisted us in identifying various species in Feihu Dong: Jiajun Zhou of Zhejiang Forest Survey, Planning and Design Company and Xingliang Wang of Guizhou Normal University and for their expertise in identifying bats; Qidi Zhu of Hebei University for her help with crickets; Ziwei Yin of Shanghai Normal University for his assistance with Staphylinidae; Jisheng Wang of Dali University for his expertise in Siphonaptera; Yejie Lin of the Institute of Zoology, Chinese Academy of Sciences for identifying spiders; Feng Lu of Shenzhen University for his help with Harvestmen; and Hengjie Huang of South China Agricultural University for assisting with Scutigeridae. Additionally, we thank Lu Qiu of Mianyang Normal University for his help with mollusks. We would like to express our sincere gratitude to the three anonymous reviewers for their valuable feedback and suggestions, which have helped us to improve the quality of our work. We would also like to thank Georgi Angelov Geshev (Imperial College London, United Kingdom) and Sergei I. Golovatch (Russian Academy of Sciences, Moscow, Russia) for their assistance in checking and validating the English of our manuscript. Finally, we extend our thanks to other people from the South China Agricultural University Biocaving Team (Haomin Yin, Xinhui Wang, Mingruo Tang, Zijun Ma, Xinyang Jia, Yi Zhao) and the team Xiangxi Cave Expedition (烧鸟, 天悟, 末路, 三皮, 华华, 小爱, 虫子, 三二, 马儿, 笑笑, 水瓶, 畅畅, 鸡腿, 伙头军, 燕子, 不二, 余老师, 小郭耳朵, 浆糊, 烧成灰) for their assistance with caving and fieldwork in Feihu Dong. Thanks also to the Speleo-Club de Paris (France) for the support of the biological team during the 1995 speleological expedition, and to the SHAG Caving Association of Besançon (France) for the final cave topographies, conducted after the 1997 speleological expedition, with the support of the authorities from Hunan (the Institute of Geology of Hunan Province and the Government of Longshan County).

**Conflicts of Interest:** The authors declare no conflict of interest.

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
