# Peer review of "Feihu Dong, a New Hotspot Cave of Subterranean Biodiversity from China"

_diversity, doi:10.3390/d15080902_

Round 1

Reviewer 1 Report

This is an interesting and well-organized work, I would suggest only a few corrections:

line 48- after "...cave biodiversity" I would add the quote from Latella L. Biodiversity:China. Encyclopedya of caves. Academic press, 2019.

line 57 - remove from "a substantial..." to "...in China" and replace with "increasing research has led to the description of a substantial number of subterranean species new to science"

line 76 - remove "there is a rich cave fauna hidden in the cave" and replace with "the cave is inhabited by abundant and diversified subterranean fauna"

line 101 - remove "cave" before Feihu Dong and add it in parentheses "(Flying Tiger CAVE in Chinese)"

lines 105 and - remove "with shaft entrances" and add "with three entrances, one of wich is a -320 m deep shaft"

Line107 - remove "underground tunnels and speleothemes"

Table 1 - is it relevant to the work? 

line 134 - remove "cave" before animals

line 228 - specify how humans may have brought the springtails

line 253 add "together" between collected and with

line 279 - remove Stygofauna and replace with "aquatic insect fauna"

line 304 - no troglobiotic but stygobiotic

lines 340 to 343 - delete all the paragraph

the sentence on line 365 seems to contradict what stated il line 358

regarding what stated in line 376:​​​​​ Drought is certainly a problem, but I would not use the presence of remains as evidence, they are often encountered in caves of different types, both tropical and temperate, and may have been flowed by laminar water or something else.

kind regards

Author Response

Dear Reviewer,

Thank you for your valuable comments and suggestions on our manuscript. We have carefully considered all of your comments and have made the necessary revisions to our manuscript. All of your suggestions have been accepted, and explanation for the table is that: "Table 1 is relevant to the work. The original cave map includes location names in French and Chinese, and we have translated these names into English in Table 1 to provide a reference for readers. This allows us to cite these locations in English throughout the paper, while still providing the original names for reference."

We have provided detailed corrections in the revised manuscript.

Thank you again for your time and effort in reviewing our work.

Best regards, 

Dr. Sunbin HUANG

Reviewer 2 Report

Dear authors,

as one of the reviewers, I am strongly supporting your efforts to publish the data on Chinese subterranean localities. As a member of speleobiological community, I have to say, that we are missing this. Therefore, I am would give a recommendation for publishing your work. However, there has to be a "but" in the story. Before publishing it, I encourage you to decide, whether you will report of cave adapted taxa (i.e. troglobionts and troglophiles) and lower the number of your taxa to 53, or you will report on the taxa found in the cave, which makes them 62. There is a big difference between the two, and the first option is more sound for your research. 

All in all, your manuscript is worth publishing, but has to be carefully edited. My comments are included as comments to the original PDF file.

Dear authors, 

please consider editing your manuscript's language once again before publishing it. I think it would be worth of it.

Author Response

Dear Reviewer,

Thank you for your valuable comments, corrections and suggestions on our manuscript. We have carefully considered all of your comments and have made the necessary revisions to our manuscript. Almost all of your corrections and suggestions have been accepted, please see the detailed corrections in the revised manuscript. 

Regarding the taxa, we would like to keep our original list (62 taxa). We have changed the words "Cave Animals" to "Animals" in appropriate places in the manuscript. This change better reflects the fact that some of the animals on our list are not all cave animals. Also, we have added new sentences to the Discussion section of the manuscript to address the issue of trogloxenes. In the sentences, we discuss the challenges of defining and classifying trogloxenes and we provide further information on the trogloxenes in Feihu Dong. We keep the parts of "Aquatic insects" and "Other insects" for the same reason. 

In "Coleoptera" part of 3.2.7. Insecta, response to point "I guess these species were known from previous explorations, so your findings are ascribed based on literature data. If yoes, please write it down.. Otherwise I do not see a connection between elitrae and the species determination.": We clarify that the elytra of the ground beetles were collected during our survey in 2023, and it is rephrased in the sentences.

Thank you again for your time and effort in reviewing our work.

Best regards, 

Dr. Sunbin HUANG

Reviewer 3 Report

The paper is well structured, the exposition in each section is concise, informative and clearly explained. The species inventory is very useful and will be more so in the future as it is completed. I encourage researchers to intensify and expand sampling, especially aquatics, and to look for taxonomists who can collaborate in species descriptions, to try to complete the lists of species that look promising. Congratulations on the photographs of the animals in situ that are very good, difficult to obtain and quite informative.

I have added small corrections or comment directly in the text (pdf).

I recommend publishing this paper with minor corrections

General Comments

The work is very interesting, it is well planned, the tables are adequated, the references are appropriate and the figures are adequeted and photographs have very good quality and are informative. 

Some specific comments:

1. The sampling methods are not very well detailed... Please expand this information.

2. Based on the data, it seems that little is known about the stygobiont fauna, and regarding the troglobiont fauna, you have estimated the level of sampling effort, and I think you should also include a note on the sampling effort carried out in the aquatic environment, even if it was a small effort... to try to understand the scarcity of stygobionts.

I hope to be able to read this work published soon!

Author Response

Dear Reviewer,

Thank you for your valuable comments and corrections on our manuscript. We have carefully considered all of your comments and have made the necessary revisions to our manuscript. All of your suggestions or corrections have been accepted, please see the detailed corrections in the revised manuscript. 

- Regarding the sampling methods: We appreciate your suggestion to expand the information on our sampling methods. Due to the scope and focus of our paper, we have added only a small amount of additional information on our sampling methods in the current version of our paper. We hope this addresses your concern and we welcome any further feedback you may have. Please check the detailed information in the revised manuscript.
- For sampling effort in the aquatic environment, we have add a note on the sampling effort in 4.4. Limitations and Prospects: "Regarding the sampling effort in the aquatic environment, we conducted very few investigations concerned about the baited traps and litter extraction in the water. The aquatic fauna of the Feihu Dong system remains poorly known, and further surveys are needed to fully understand the diversity and ecology of this unique ecosystem."

Thank you again for your time and effort in reviewing our work.

Best regards, 

Dr. Sunbin HUANG
